# Improving Shift Invariance in Convolutional Neural Networks with Translation Invariant Polyphase Sampling

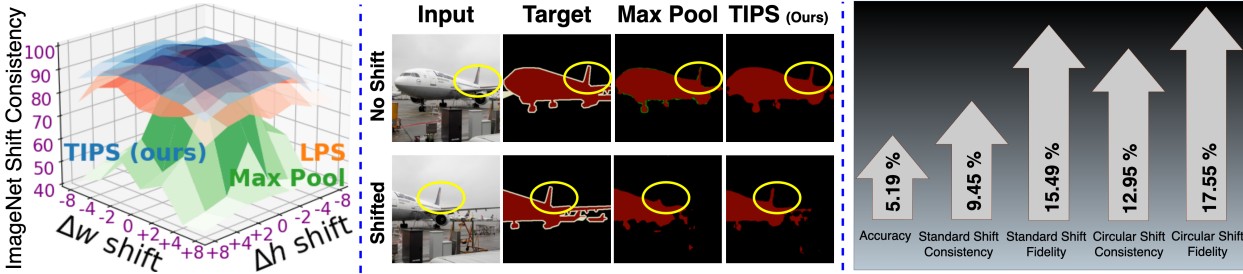

Figure 1: This paper develops Translation-Invariant Polyphase Sampling (TIPS), a pooling operator that improves shift invariance of CNNs. **Left:** For image classification, shift consistency of TIPS (blue) is significantly higher than LPS (orange), and MaxPool (green) especially at higher degrees of pixel shift; **middle:** semantic segmentation networks with a TIPS pooling layer leads to greater shift equivariance than previous methods; **right:** TIPS results in consistent and architecture-agnostic improvements in accuracy and four measures of shift invariance, across multiple image classification and segmentation benchmarks.

## Abstract

Downsampling operators break the shift invariance of convolutional neural networks (CNNs) and this affects the robustness of features learned by CNNs when dealing with even small pixel-level shift. Through a large-scale correlation analysis framework, we study shift invariance of CNNs by inspecting existing downsampling operators in terms of their *maximum-sampling bias (MSB)*, and find that MSB is negatively correlated with shift invariance. Based on this crucial insight, we propose a learnable pooling operator called *Translation Invariant Polyphase Sampling (TIPS)* and two regularizations on the intermediate feature maps of TIPS to reduce MSB and learn translation-invariant representations. TIPS can be integrated into any CNN and can be trained end-to-end with marginal computational overhead. Our experiments demonstrate that TIPS results in consistent performance gains [1] in terms of accuracy, shift consistency, and shift fidelity on multiple benchmarks for image classification and semantic segmentation compared to previous methods and also leads to improvements in adversarial and distributional robustness. TIPS results in the lowest MSB compared to all previous methods, thus explaining our strong empirical results.

## 1 Introduction

Shift invariance is an ideal property for visual recognition models and necessitates that predictions remain invariant to small pixel-level shifts in input images. Shifting an image by a few pixels horizontally and/or vertically should not affect the category predicted by an image classifier such as a convolutional neural network (CNN). Figure 2 depicts three scenarios where an input $x$ undergoes a transformation $g$ before being fed into a model $f$ to generate a prediction $\hat{y} = f(g(x)) = g'(f(x))$: shift equivariance, shift non-invariance, and shift invariance. If $g' = g$, then $f$ is $g$-equivariant and if $g' = I$ then $f$ is $g$-invariant.

---

[1]Code is available at https://anonymous.4open.science/r/TIPS_review-8FDE/

Shift-invariance is desirable for image classification to ensure that categorical outputs are invariant to pixel shift, and shift-equivariance is desirable for semantic segmentation to ensure that pixel-shift in the image results in equivalent shift in segmentation. Recent studies have also found shift invariant visual recognition models to be more robust on *out-of-distribution* testing and adversarial attacks, therefore improving shift invariance in CNNs is consequential.

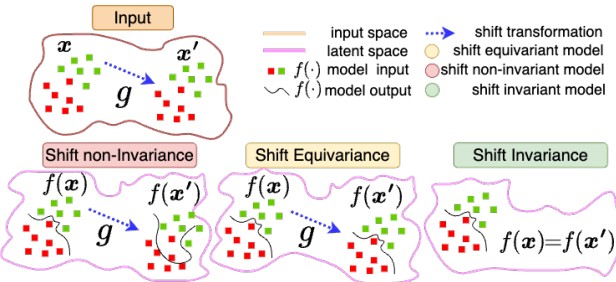

Figure 2: An illustration of shift equivariance, non-invariance, and invariance. Invariant models map shifted, non-shifted inputs to identical outputs, while equivariant models mirror the input shift in outputs.

Although individual convolution operations in CNNs are shift-equivariant (Fukushima, 1980; Le-Cun et al., 1989; 2015), recent studies (Zhang, 2019; Azulay & Weiss, 2019; Zou et al., 2023) reveal that conventional pooling operators in CNNs such as max pooling, average pooling, and strided convolution break shift invariance by violating the Nyquist sampling theorem (Nyquist, 1928) and aliasing high-frequency signals, and impact model prediction at pooling boundaries under small shift transformations. Pooling techniques such as Max pooling, Avg Pooling and, Adaptive Polyphase Sampling (APS) (Chaman & Dokmanic, 2021) are subject to activation strength within pooling windows, i.e. the maximum feature activation in a pooling window influences the pooled value. While pooling methods such as Adaptive Polyphase Sampling (APS) (Chaman & Dokmanic, 2021) and Learnable Polyphase Sampling (LPS) (Rojas-Gomez et al., 2022) work well for circular shift, empirical evidence (Rojas-Gomez et al., 2023; Ding et al., 2023; Zhong et al., 2023) suggests lack of robustness for standard shifts. This observation raises an important question: while sampling the strongest signal works well for downstream tasks, does it affect the network's performance under pixel shift?

In this work, we study the correlation between shift invariance and the tendency to downsample strongest features by introducing the concept of Maximum-Sampling Bias (MSB). We observe a strong negative correlation between MSB and shift invariance, i.e. models with higher MSB are the least shift invariant. Based on insights from our large-scale correlation study, we design a novel pooling operator called Translation Invariant Polyphase Sampling (TIPS) that discourages MSB and improves invariance under shift transformations. To further improve visual recognition performance and shift invariance, we introduce two loss functions: $\mathcal{L}_{FM}$ – to discourage known *failure modes* of shift invariance and $\mathcal{L}_{undo}$ – to learn to *undo* standard shift. In real world scenarios, standard shifts are more like to occur than circular shift; however current literature largely focuses on circular shift invariance. We show that the TIPS pooling operator and regularization improves shift invariance on both circular shift and standard shift.

Our contributions and findings are summarized below and results are highlighted in Figure 1.

- We identify maximum-sampling bias (MSB) as a factor that hurts the shift invariance of existing pooling methods in CNNs.
- We propose a learnable pooling method called Translation Invariant Polyphase Sampling (TIPS) and two regularizations called $\mathcal{L}_{FM}$ and $\mathcal{L}_{undo}$ to improve shift invariance and discourage MSB when training visual recognition models.
- We demonstrate that this approach consistently improves robustness under shift transformation on multiple image classification and semantic segmentation benchmarks, outperforming data augmentation and contrastive learning strategies, and resulting in state-of-the-art performance in terms of accuracy, shift consistency, and shift fidelity under standard and circular shift transformations, while operating at a small computational overhead.
- When tested on adversarial attacks, patch attacks, and natural corruption of images, models trained with TIPS exhibit greater robustness than previous shift-invariant pooling operators.

## 2 Related Work

**Robustness of CNNs** has been examined under different types of input perturbations and transformations such as rotation, reflection and scaling (Cohen & Welling, 2016; Poulenard et al., 2019), geometric transformations (Liu et al., 2019), affine transformations (Engstrom et al., 2019), domain shift (Venkateswara et al., 2017), attribute shift (Gokhale et al., 2021), adversarial attacks and perturbations (Agarwal et al., 2020; Zhang et al., 2021), and natural corruptions (Hendrycks & Dietterich, 2018). Distributional robustness of CNNs has been explored through various approaches including static, random, or learned data augmentation (Hendrycks et al., 2019; Xu et al., 2020; Gokhale et al., 2023), contrastive learning (Khosla et al., 2020), and Bayesian approaches (Cheng et al., 2023).

**Dense Sampling and Anti-aliasing.** Conventional sliding window downsampling in computer vision algorithms (Fukushima, 1980; Lowe, 1999) is typically applied with stride that is bigger than 1 which breaks shift equivariance (Simoncelli et al., 1992). Shift invariance can be improved through dense sampling (Leung & Malik, 2001) with dilated convolutions (Yu et al., 2017) with susceptibility to griding artifacts. Zhang (2019) suggest BlurPool to enhance SI through anti-aliasing before downsampling, whereas Zou et al. (2020) propose DDAC, to learn low pass anti-aliasing filter.

**Polyphase Sampling.** Recent works such as APS (Chaman & Dokmanic, 2021) and LPS (Rojas-Gomez et al., 2022) use polyphase sampling to meet the Nyquist sampling theorem (Nyquist, 1928) and permutation invariance which provides robustness against circular shifts. APS enhances shift invariance by sampling the highest energy polyphase index ($\ell_p$ norm) while LPS learns the sampling. LPS being sensitive to gumble softmax temperature can sample polyphases to maximize downstream objective which does not consider shift invariance unless training data is shift-augmented. Evidence suggests that although polyphase sampling methods can improve shift invariance for circular shift, they still struggle to deal with standard shift. The focus of this study is to improve robustness of CNNs against both standard and circular shifts, which constitute significant aspects of model evaluation.

## 3 Translation Invariant Polyphase Sampling

In this section, we discuss the design of the TIPS layer and the workflow for training CNNs with TIPS. Let $X \in \mathbb{R}^{c \times h \times w}$ be a ReLU-activated input feature map, where $c$, $h$, $w$ denote the number of channels, height, and width of feature maps. A pooling layer with stride $s$, downsamples $X$ into $\hat{X}$ where $\hat{X} \in \mathbb{R}^{+c \times h/s \times w/s}$.

### 3.1 TIPS: A Learnable Pooling Layer

TIPS *learns* to sample polyphase decompositions of input feature maps $X$ using two branches. In the first branch, polyphase components of $X$ with stride $s$ are computed similar to Chaman & Dokmanic (2021):

$$\text{poly}_{is+j}(X) = X[k, sn_1 + i, sn_2 + j], \qquad \forall \; i,j \in \mathbb{Z}_0^{s-1}; k \in \mathbb{Z}_0^{c-1}; n_1 \in \mathbb{Z}_0^{\left\lceil \frac{h}{s} \right\rceil}; n_2 \in \mathbb{Z}_0^{\left\lceil \frac{w}{s} \right\rceil}. \tag{1}$$

Note that polyphase sampling can also be achieved by a strided convolution with a $s \times s$ kernel equal to 1 at index $(i,j)$ and 0 elsewhere. A visualization of polyphase sampling with $s = 2$ is shown in Figure 3. The second branch of TIPS is a small fully convolutional function $f_\theta : X \to \tau$ that learns the mixing coefficients $\tau \in [0,1]^{c \times s^2}$. $\psi()$ is the first $3 \times 3$ convolutional layer followed by ReLU activation as shown in Figure 3; thus $\psi(X)$ represents intermediate feature maps of $X$ in $f_\theta()$. All operations in $f_\theta$ are shift invariant, including Global Average Pooling (GAP) (He et al., 2016b). The output of the TIPS layer is computed as a weighted linear combination of the polyphase components:

$$\hat{X} = \sum_{i,j} \tau_{is+j} \, \text{poly}_{is+j}(X) \tag{2}$$

**Regularizing TIPS to Discourage Known Failure Modes of Shift Invariance.** Chaman & Dokmanic (2021); Rojas-Gomez et al. (2022) have shown that having extremely skewed mixing coefficients (*e.g.*

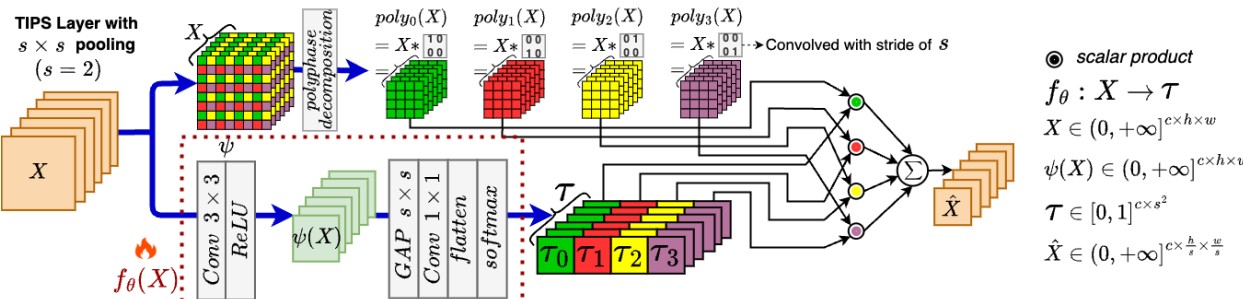

Figure 3: TIPS downsamples ReLU activated intermediate feature map $X$ into $\hat{X}$ with stride of $s$. From input feature map $X$, TIPS learns polyphase mixing coefficients $\tau$ using a small fully convolutional function $f_\theta$. The polyphase decomposition on input feature map $X$ results in $\text{poly}_i$ which are then mixed as a weighted linear combination with $\tau$ (Equation 2) to compute $\hat{X}$.

$\tau=\{0,1,0,0\}$ for $s=2$) is not robust against standard shift. TIPS with uniform mixing coefficients and LPS with higher softmax temperature (*e.g.* $\tau=\{0.25, 0.25, 0.25, 0.25\}$ for $s=2$) is identical to average pooling, which has been shown to hurt shift invariance (Zhang, 2019; Zou et al., 2020). Based on these observations, we introduce a regularization on the mixing coefficients in TIPS to discourage known failure modes. In $\mathcal{L}_{FM}$, the first term discourages skewed $\tau$ and the second term discourages uniform $\tau$.

$$\mathcal{L}_{FM} = (\|\tau\|_2 - 1) + (1 - s^2 \|\tau\|_2) = (1 - s^2)\|\tau\|_2. \tag{3}$$

**Learning to Undo Standard Shift.** Although prior work (Chaman & Dokmanic, 2021; Rojas-Gomez et al., 2022; 2023; Ding et al., 2023; Zhong et al., 2023) has shown the benefits of using polyphase sampling to counter circular shifts, there is still a performance degradation with standard shifts due to information loss beyond the pooling boundary. To improve robustness against standard shift, we shift the input feature map with a random amount of vertical and horizontal standard shift sampled from uniform distribution $U(0, \frac{h}{10})$ and $U(0, \frac{w}{10})$ respectively to obtain a shifted $X^t$. We then regularize training by setting up the objective of undoing this shift between $\psi(X)$ and the shifted $X^t$, via the additional loss term $\mathcal{L}_{undo}$:

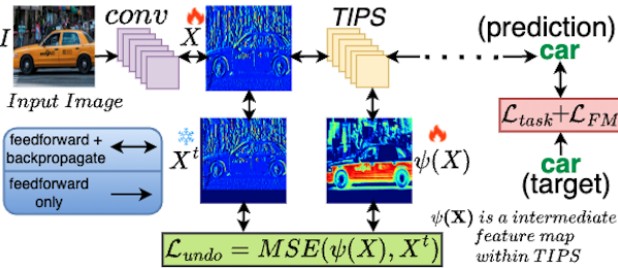

Figure 4: The *end-to-end* training pipeline with TIPS, Translation invariant regularization $\mathcal{L}_{undo}$, regularization to discourage known failure modes of shift invariance $\mathcal{L}_{FM}$, and downstream task loss $\mathcal{L}_{task}$.

$$\mathcal{L}_{undo} = \mathbb{E}_{h' \in h}\mathbb{E}_{w' \in w}[X^t_{h',w'} - \psi(X_{h',w'})]^2 \tag{4}$$

## 3.2  Training CNNs with TIPS

Let $N$ be the number of training epochs. For the first $\epsilon N$ epochs, we train only with the task loss $\mathcal{L}_{task}$ and the regularization to discourage failure models $\mathcal{L}_{FM}$. For subsequent epochs, the undo regularization $\mathcal{L}_{undo}$ is introduced. The final training loss is the Lagrangian (with $\alpha \in [0,1]$):

$$\mathcal{L} = \begin{cases} (1-\alpha)\mathcal{L}_{task} + \mathcal{L}_{FM} & \text{for } epoch < \epsilon N \\ (1-\alpha)\mathcal{L}_{task} + \alpha\mathcal{L}_{undo} + \mathcal{L}_{FM} & \text{otherwise.} \end{cases} \tag{5}$$

$\psi(X)$ contains a $3 \times 3$ convolution layer, followed by ReLU. This is followed by global average pooling, $1 \times 1$ convolution, flattening, and a softmax operation to obtain $\tau$ as shown in Figure 3. Weights of $f_\theta$ are initialized using the using Kaiming normal approach (He et al., 2016a).

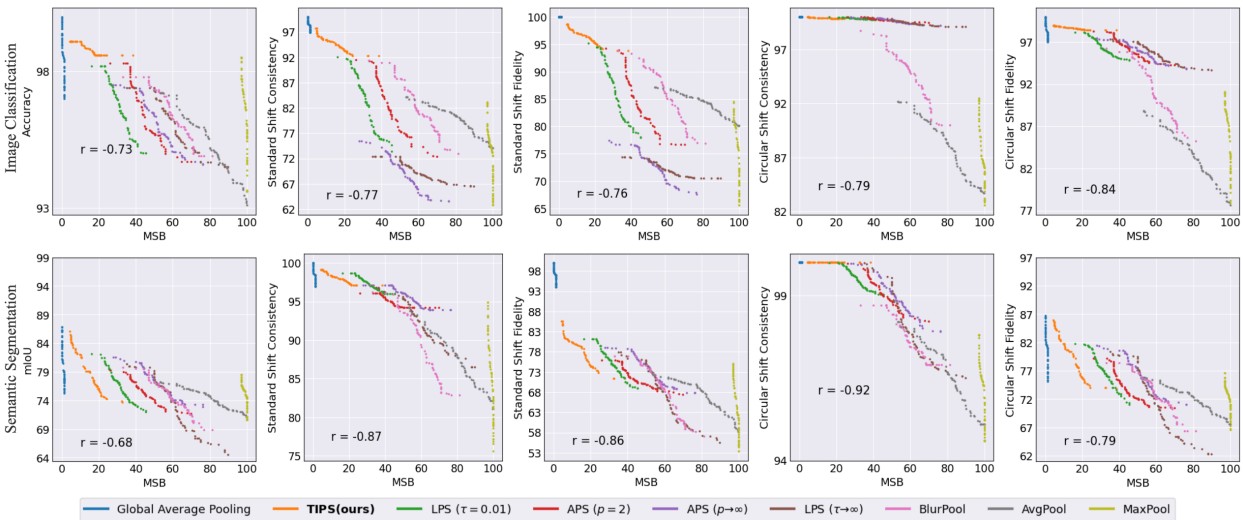

Figure 5: Our large-scale study of shift invariance of CNN-based models for image classification and semantic segmentation, with multiple CNN architectures, datasets, and pooling methods, shows a strong negative correlation between each evaluation metric and MSB (%), as indicated by the Pearson correlation coefficient (r). Linear clusters with negative correlation are also observed for points belonging to each pooling method.

## 4 Maximum-Sampling Bias and its Correlation with Shift Invariance

In this section, we setup a framework to study shift invariance in CNNs, by defining maximum-sampling bias (MSB). We show that MSB is a common preference exhibited by both conventional pooling operators and those designed to improve shift invariance and through a large-scale analysis, we show that MSB is negatively correlated with shift invariance.

**Definition of MSB.** Existing pooling operators exhibit a common tendency to propagate signals based on activation strength. We denote this phenomenon as maximum-sampling bias (MSB), defined as the fraction of window locations for which the the maximum signal value is sampled. MSB quantifies the *bias* of a pooling operator to select and propagate the maximum value of the signal; a higher MSB indicates a higher probability of maximum signal values being selected during pooling. Let $p()$ be a pooling operator in a convolutional neural network and $s$ be the downsampling factor; for example, $s=2$ for a max-pooling operator with window size $2 \times 2$. Let $X \in \mathbb{R}^{h \times w}$ be the 2-dimensional input to a pooling layer. Applying a pooling operator $p()$ on $X$ with downsampling factor $s$ results in an output $\hat{X} = p(X) \in \mathbb{R}^{\frac{h}{s} \times \frac{w}{s}}$.

It is trivial to see that $MSB = 1$ for max-pool, as max-pool by definition always selects the maximum signal, $\hat{X}[i,j] = \max_{m,n} X[is+m, js+n] \ \forall(i,j)$. Average pooling produces $X[i,j] = \mathbb{E}_{m,n} x[is+m, js+n]$ and is equivalent to max-pooling if all values within the window are identical. For all other cases, the average value is sampled, which is necessarily less than the maximum and thus $MSB \leq 1$. APS pooling (Chaman & Dokmanic, 2021) samples the polyphase component of $X$ with maximum $\ell_p$ norm; $\hat{X}[i,j] = \max_{is+j} \{\|poly_j(x)\|_p\}_{i,j=0}^{s-1}$. As the polyphase function $poly()$ in APS and LPS (Rojas-Gomez et al., 2022) is a monotonic function, it also exhibits a preference for sampling larger signals in the pooling window.

From the above definition, we can see that existing pooling operators implicitly prefer selecting larger elements in the pooling window. We investigate whether this preference (or bias) towards maximum-sampling is linked to shift invariance, by conducting a large-scale analysis of the correlation between MSB and shift invariance on a number of visual recognition benchmarks with multiple CNN architectures and pooling methods. Our work is the first to identify maximum-sampling bias and its connection to shift invariance.

**(a) CIFAR-10**

| | Method | Acc. | Standard Shift | | Circular Shift | |
| --- | --- | --- | --- | --- | --- | --- |
| | | | Consistency | Fidelity | Consistency | Fidelity |
| CNN (ResNet-18) | MaxPool | 91.43±0.04 | 87.43±0.05 | 79.94±0.05 | 90.18±0.03 | 82.45±0.08 |
| | APS | 94.02±0.07 | 92.89±0.08 | 87.33±0.05 | 100.00±0.00 | 94.02±0.07 |
| | LPS | 94.45±0.05 | 93.11±0.07 | 87.94±0.03 | 100.00±0.00 | 94.45±0.05 |
| | **TIPS** | 95.75±0.11 | 98.38±0.37 | 94.20±0.08 | 100.00±0.00 | 95.75±0.11 |
| | BlurPool (LPF-5) | 94.29±0.11 | 91.04±0.09 | 85.84±0.12 | 98.27±0.11 | 92.66±0.07 |
| | APS (LPF-5) | 94.44±0.09 | 93.25±0.13 | 88.06±0.17 | 100.00±0.00 | 94.44±0.09 |
| | LPS (LPF-5) | 95.17±0.12 | 94.87±0.08 | 90.09±0.15 | 100.00±0.00 | 95.17±0.12 |
| | **TIPS (LPF-5)** | 96.05±0.13 | 98.65±0.11 | 94.75±0.10 | 100.00±0.00 | 96.05±0.13 |
| ViT | ViT-B/16 (I21k) | 98.89±0.04 | 82.34±0.07 | 81.43±0.05 | 83.79±0.15 | 82.86±0.12 |
| | ViT-L/16 (I21k) | 99.15±0.02 | 82.72±0.09 | 82.01±0.08 | 84.41±0.11 | 83.69±0.06 |
| | Swin-B (I21k) | 99.22±0.03 | 83.19±0.07 | 82.54±0.05 | 84.05±0.04 | 83.40±0.04 |

**(b) CIFAR-100**

| | Method | Acc. | Standard Shift | | Circular Shift | |
| --- | --- | --- | --- | --- | --- | --- |
| | | | Consistency | Fidelity | Consistency | Fidelity |
| CNN (ResNet-34) | MaxPool | 88.38±0.04 | 90.25±0.07 | 79.76±0.13 | 88.21±0.08 | 77.96±1.05 |
| | APS | 88.49±0.12 | 93.54±0.11 | 82.77±0.09 | 100.00±0.00 | 88.49±0.12 |
| | LPS | 87.62±0.07 | 92.73±0.18 | 81.25±0.15 | 100.00±0.00 | 87.62±0.07 |
| | **TIPS** | 91.86±0.03 | 95.77±0.04 | 87.97±0.12 | 100.00±0.00 | 91.86±0.03 |
| | BlurPool (LPF-5) | 87.79±0.11 | 92.65±0.14 | 81.34±0.14 | 95.39±0.10 | 73.81±0.14 |
| | APS (LPF-5) | 88.57±0.07 | 93.97±0.04 | 83.20±0.02 | 100.00±0.00 | 88.57±0.07 |
| | LPS (LPF-5) | 88.79±0.12 | 93.41±0.08 | 83.00±0.11 | 100.00±0.00 | 88.79±0.12 |
| | **TIPS (LPF-5)** | 92.34±0.09 | 95.96±0.11 | 88.61±0.07 | 100.00±0.00 | 92.34±0.09 |
| ViT | ViT-B/16 (I21k) | 91.54±0.07 | 87.25±0.08 | 79.87±0.10 | 82.39±0.04 | 75.42±0.06 |
| | ViT-L/16 (I21k) | 93.39±0.05 | 87.11±0.18 | 81.35±0.15 | 81.49±0.07 | 76.11±0.11 |
| | Swin-B (I21k) | 93.78±0.03 | 87.34±0.06 | 81.91±0.17 | 83.57±0.11 | 78.37±0.13 |

Table 1: Image classification performance on CIFAR-10 and CIFAR-100 averaged over five trials.

**Negative Correlation between MSB and Shift Invariance.** To understand how MSB affects shift invariance in CNNs we evaluated 576 models across different architectures, datasets, and pooling methods [2] and conducted a correlation study as shown in Figure 5 with MSB on x-axis and performance metrics on the y-axis for both image classification and semantic segmentation. A strong negative correlation is observed between MSB and shift consistency and fidelity (discussed in the next subsection), and surprisingly also with downstream task performance (accuracy, mIoU). In all scenarios, when MSB decreases, shift invariance and downstream performance improves. Figure 5 further depicts that circular consistency is more negatively correlated with MSB than standard consistency for both tasks. Linear relationships are also observed for points corresponding to specific pooling methods across architectures and datasets. Using Global Average Pooling (GAP) (He et al., 2016a) before classification layer with no spatial downsampling of the intermediate feature maps leads to additional computational expense since there are more feature grids to convolve. While this design choice helps improving shift invariance, the computational expense for additional convolution operations renders such designs impractical (see Appendix Table 11 for computational costs). TIPS achieves high shift invariance with marginal computational overhead.

# 5 Experiments

We perform experiments on multiple benchmarks for both image classification and semantic segmentation. For image classification, we evaluate shift invariance while for semantic segmentation we evaluate shift equivariance; following conventions used in prior work, this is also referred to as "shift invariance" in the results. For both classification and segmentation, we avoid using pre-trained CNNs since the pre-training step uses strided convolution and maxpool.

## 5.1 Image Classification Experiments

**Datasets and Baselines.** We benchmark the performance of TIPS and prior work on six image classification datasets: CIFAR-10, 100 (Krizhevsky, 2009), Food-101 (Bossard et al., 2014), Oxford-102 (Nilsback & Zisserman, 2008), Tiny ImageNet (Le & Yang, 2015), and ImageNet (Krizhevsky et al., 2012). Our baselines include MaxPool, APS ($p$=2), and LPS ($\tau$=0.23), as well as BlurPool, APS, and LPS with anti-aliasing using $n \times n$ Gaussian low-pass filter (LPF-5). We also compare with three Vision Transformer (ViT) architectures: ViT-B/16, ViT-L-16 (Dosovitskiy et al., 2020), and Swin-B (Liu et al., 2021) which are pre-trained on the larger ImageNet-21k dataset (Deng et al., 2009).

**Hyperparameters.** For TIPS, we choose $\epsilon = 0.4$ and $\alpha = 0.35$ in Equation 5. All models are trained using an SGD optimizer with initial learning rate 0.05, momentum 0.9, and weight decay 1e-4 with early stopping. No models in our experiments were trained on shifted images. For each dataset-backbone pair, for fair comparison, TIPS and all baselines are trained with identical hyperparameters.

---

[2]The appendix has details about architectures, datasets, pooling methods, and hyperparameters for the all experiments.

|  | Method | Acc. | Standard Shift | | Circular Shift | |
|---|---|---|---|---|---|---|
|  |  |  | Consistency | Fidelity | Consistency | Fidelity |
| CNN (ResNet-50) | MaxPool | $92.96_{\pm0.08}$ | $82.13_{\pm0.57}$ | $76.18_{\pm0.07}$ | $83.61_{\pm0.12}$ | $77.72_{\pm0.05}$ |
| | APS | $94.68_{\pm0.11}$ | $91.34_{\pm0.04}$ | $86.48_{\pm0.13}$ | $100.00_{\pm0.00}$ | $94.68_{\pm0.11}$ |
| | LPS | $94.71_{\pm0.02}$ | $92.41_{\pm0.03}$ | $87.52_{\pm0.11}$ | $99.48_{\pm0.11}$ | $94.22_{\pm0.05}$ |
| | **TIPS** | $95.63_{\pm0.15}$ | $95.02_{\pm0.09}$ | $90.87_{\pm1.08}$ | $100.00_{\pm0.00}$ | $95.63_{\pm0.15}$ |
| | BlurPool (LPF-5) | $93.77_{\pm0.03}$ | $88.18_{\pm0.17}$ | $82.69_{\pm1.08}$ | $93.49_{\pm0.13}$ | $87.67_{\pm0.03}$ |
| | APS (LPF-5) | $94.07_{\pm0.13}$ | $92.51_{\pm0.06}$ | $87.03_{\pm0.20}$ | $100.00_{\pm0.00}$ | $94.07_{\pm0.13}$ |
| | LPS (LPF-5) | $95.62_{\pm0.07}$ | $94.10_{\pm0.07}$ | $89.99_{\pm0.19}$ | $100.00_{\pm0.00}$ | $95.62_{\pm0.07}$ |
| | **TIPS (LPF-5)** | $96.42_{\pm0.16}$ | $95.50_{\pm0.13}$ | $92.08_{\pm0.19}$ | $100.00_{\pm0.00}$ | $96.42_{\pm0.16}$ |
| ViT | ViT-B/16 (I21k) | $96.88_{\pm0.13}$ | $81.45_{\pm0.04}$ | $78.91_{\pm0.15}$ | $78.39_{\pm0.12}$ | $75.94_{\pm0.12}$ |
| | ViT-L/16 (I21k) | $97.00_{\pm0.03}$ | $81.84_{\pm0.11}$ | $79.38_{\pm0.06}$ | $78.06_{\pm0.18}$ | $75.72_{\pm0.17}$ |
| | Swin-B (I21k) | $97.49_{\pm0.05}$ | $82.85_{\pm0.14}$ | $80.77_{\pm0.09}$ | $78.05_{\pm0.02}$ | $76.10_{\pm0.08}$ |

(a) Food-101

|  | Method | Acc. | Standard Shift | | Circular Shift | |
|---|---|---|---|---|---|---|
|  |  |  | Consistency | Fidelity | Consistency | Fidelity |
| CNN (ResNet-50) | MaxPool | $93.48_{\pm0.15}$ | $85.63_{\pm0.11}$ | $80.05_{\pm0.17}$ | $89.38_{\pm0.17}$ | $83.55_{\pm0.12}$ |
| | APS | $94.68_{\pm0.03}$ | $92.47_{\pm0.05}$ | $87.55_{\pm1.09}$ | $100.00_{\pm0.00}$ | $94.68_{\pm0.03}$ |
| | LPS | $95.31_{\pm0.08}$ | $93.63_{\pm0.17}$ | $89.24_{\pm0.11}$ | $100.00_{\pm0.00}$ | $95.31_{\pm0.08}$ |
| | **TIPS** | $97.18_{\pm0.16}$ | $95.78_{\pm0.03}$ | $93.08_{\pm0.16}$ | $100.00_{\pm0.00}$ | $97.18_{\pm0.06}$ |
| | BlurPool (LPF-5) | $92.71_{\pm0.08}$ | $90.32_{\pm0.13}$ | $83.74_{\pm0.05}$ | $94.07_{\pm0.13}$ | $87.21_{\pm0.08}$ |
| | APS (LPF-5) | $94.71_{\pm0.11}$ | $93.00_{\pm0.08}$ | $88.09_{\pm0.14}$ | $100.00_{\pm0.00}$ | $94.71_{\pm0.11}$ |
| | LPS (LPF-5) | $96.28_{\pm0.05}$ | $94.33_{\pm0.06}$ | $90.82_{\pm0.09}$ | $100.00_{\pm0.00}$ | $96.28_{\pm0.05}$ |
| | **TIPS (LPF-5)** | $97.62_{\pm0.11}$ | $96.51_{\pm0.14}$ | $94.21_{\pm0.14}$ | $100.00_{\pm0.00}$ | $97.62_{\pm0.11}$ |
| ViT | ViT-B/16 (I21k) | $99.33_{\pm0.05}$ | $88.47_{\pm0.04}$ | $87.88_{\pm0.08}$ | $82.24_{\pm0.03}$ | $81.69_{\pm0.06}$ |
| | ViT-L/16 (I21k) | $99.59_{\pm0.03}$ | $87.25_{\pm0.09}$ | $86.89_{\pm0.18}$ | $82.39_{\pm0.13}$ | $82.05_{\pm0.03}$ |
| | Swin-B (I21k) | $99.68_{\pm0.02}$ | $87.06_{\pm0.16}$ | $80.16_{\pm0.07}$ | $83.57_{\pm0.11}$ | $83.30_{\pm0.05}$ |

(b) Oxford-102

Table 2: Image classification performance on Food-101 and Oxford-102 datasets averaged over five trials.

|  | Method | Acc. | Standard Shift | | Circular Shift | |
|---|---|---|---|---|---|---|
|  |  |  | Consistency | Fidelity | Consistency | Fidelity |
| CNN (ResNet-101) | MaxPool | $78.54_{\pm0.22}$ | $88.45_{\pm0.15}$ | $69.47_{\pm0.14}$ | $92.82_{\pm.14}$ | $79.20_{\pm0.15}$ |
| | APS | $83.01_{\pm0.08}$ | $91.37_{\pm0.06}$ | $75.85_{\pm0.04}$ | $100.00_{\pm0.00}$ | $83.01_{\pm0.08}$ |
| | LPS | $85.67_{\pm0.18}$ | $92.95_{\pm0.04}$ | $79.63_{\pm0.06}$ | $100.00_{\pm0.00}$ | $85.67_{\pm0.18}$ |
| | **TIPS** | $86.78_{\pm0.19}$ | $94.27_{\pm0.14}$ | $81.80_{\pm0.15}$ | $100.00_{\pm0.00}$ | $86.78_{\pm0.19}$ |
| | BlurPool (LPF-5) | $82.83_{\pm0.13}$ | $90.81_{\pm0.17}$ | $75.22_{\pm0.12}$ | $95.87_{\pm0.19}$ | $79.41_{\pm1.12}$ |
| | APS (LPF-5) | $83.52_{\pm0.03}$ | $92.00_{\pm0.20}$ | $76.84_{\pm0.11}$ | $100.00_{\pm0.00}$ | $83.52_{\pm0.03}$ |
| | LPS (LPF-5) | $86.74_{\pm0.09}$ | $93.38_{\pm0.17}$ | $80.99_{\pm0.04}$ | $100.00_{\pm0.00}$ | $86.74_{\pm0.09}$ |
| | **TIPS (LPF-5)** | $86.91_{\pm0.13}$ | $94.55_{\pm0.06}$ | $88.20_{\pm0.07}$ | $100.00_{\pm0.00}$ | $86.91_{\pm0.13}$ |
| ViT | ViT-B/16 (I21k) | $89.34_{\pm0.06}$ | $73.47_{\pm0.03}$ | $65.64_{\pm0.11}$ | $72.94_{\pm0.19}$ | $65.16_{\pm0.08}$ |
| | ViT-L/16 (I21k) | $90.75_{\pm0.15}$ | $74.39_{\pm0.16}$ | $67.51_{\pm0.14}$ | $73.85_{\pm0.06}$ | $67.02_{\pm0.19}$ |
| | Swin-B (I21k) | $91.19_{\pm0.04}$ | $72.14_{\pm0.15}$ | $65.78_{\pm0.17}$ | $75.49_{\pm0.05}$ | $68.84_{\pm0.10}$ |

(a) TinyImageNet

|  | Method | Acc. | Standard Shift | | Circular Shift | |
|---|---|---|---|---|---|---|
|  |  |  | Consistency | Fidelity | Consistency | Fidelity |
| CNN (ResNet-101) | MaxPool | $76.31_{\pm0.18}$ | $89.05_{\pm0.19}$ | $67.05_{\pm0.06}$ | $87.56_{\pm0.13}$ | $66.82_{\pm0.17}$ |
| | APS | $76.07_{\pm0.15}$ | $90.95_{\pm0.13}$ | $69.19_{\pm0.13}$ | $100.00_{\pm0.00}$ | $76.07_{\pm0.15}$ |
| | LPS | $78.29_{\pm0.14}$ | $91.74_{\pm0.03}$ | $71.82_{\pm0.13}$ | $100.00_{\pm0.00}$ | $78.29_{\pm0.14}$ |
| | **TIPS** | $80.24_{\pm0.09}$ | $92.87_{\pm0.08}$ | $74.52_{\pm0.18}$ | $100.00_{\pm0.00}$ | $80.24_{\pm0.09}$ |
| | BlurPool (LPF-5) | $76.33_{\pm0.08}$ | $90.70_{\pm0.14}$ | $69.23_{\pm0.15}$ | $90.55_{\pm0.17}$ | $69.12_{\pm0.19}$ |
| | APS (LPF-5) | $76.49_{\pm0.08}$ | $91.23_{\pm0.17}$ | $69.78_{\pm0.05}$ | $99.98_{\pm0.00}$ | $76.41_{\pm0.06}$ |
| | LPS (LPF-5) | $78.31_{\pm0.05}$ | $92.49_{\pm0.15}$ | $72.43_{\pm0.04}$ | $100.00_{\pm0.00}$ | $78.31_{\pm0.05}$ |
| | **TIPS (LPF-5)** | $81.36_{\pm0.10}$ | $93.11_{\pm0.03}$ | $75.75_{\pm0.14}$ | $100.00_{\pm0.00}$ | $81.36_{\pm0.10}$ |
| ViT | ViT-B/16 (I21k) | $83.89_{\pm0.07}$ | $84.38_{\pm0.05}$ | $70.79_{\pm0.27}$ | $81.03_{\pm0.11}$ | $67.98_{\pm0.19}$ |
| | ViT-L/16 (I21k) | $85.06_{\pm0.02}$ | $83.19_{\pm0.12}$ | $70.76_{\pm0.17}$ | $81.64_{\pm0.15}$ | $69.44_{\pm0.14}$ |
| | Swin-B (I21k) | $85.16_{\pm0.05}$ | $85.24_{\pm0.19}$ | $72.59_{\pm0.05}$ | $82.79_{\pm0.08}$ | $70.50_{\pm0.18}$ |

(b) ImageNet

Table 3: Image classification performance on TinyImageNet and ImageNet averaged over five trials.

**Evaluation Metrics.** In addition to reporting classification accuracy on the unshifted test set, we use the *consistency* definition from Zou et al. (2020) which compares the predictions for two shifted images. However, as *consistency* does not consider the ground truth label ($y$) for evaluation, we introduce *fidelity* as a new metric. Note: $x^{h_1,w_1}$ denotes image $x$ shifted by $h \sim U(0, h/8)$ vertically and $w \sim U(0, w/8)$ horizontally.

$$\text{Consistency} = \mathbb{E}_{x} \mathbb{E}_{(h_1,w_1),(h_2,w_2)} \mathbb{1}[f(x^{h_1,w_1}) = f(x^{h_2,w_2})]. \tag{6}$$

$$\text{Fidelity} = \mathbb{E}_{x} \mathbb{E}_{(h_1,w_1),(h_2,w_2)} \mathbb{1}[y = f(x^{h_1,w_1}) = f(x^{h_2,w_2})]. \tag{7}$$

**Results.** Tables 1, 2, and 3 show strong dataset- and backbone-agnostic evidence for the efficacy of TIPS in terms of accuracy and shift invariance for both standard shift and circular shift. TIPS results in large gains in consistency and fidelity on standard shift, which was a challenge for prior work. It is important to note that TIPS with LPF-5 also improves upon prior work that uses LPF-5 anti-aliasing. For ViTs, shift invariance performance is inferior to CNNS, even though they consistently achieve higher accuracy. ViT architectures - despite being pre-trained on a very large scale dataset ImageNet21k (I21k) cannot improve shift invariance which depicts that large-scale pre-training has no implications on shift invariace. While CNNs in general perform better on circular shift than standard shift, there is no such clear trend for ViT – for example, ViTs are more robust on standard shift for Oxford-102 and Tiny ImageNet and more robust on circular shift for the other four datasets.

## 5.2 Semantic Segmentation Experiments

**Datasets and Baselines.** We use the following datasets: PASCAL VOC 2012 (Everingham et al., 2010), Cityscapes (Cordts et al., 2016), Kvasir (Jha et al., 2020), and CVC-ClinicDB (Bernal et al., 2015). Our baselines include MaxPool, APS, LPS, BlurPool (LPF-3), and DDAC (groups $g$=8, LPF-3). BlurPool and DDAC (Zou et al., 2020) perform antialiasing by either using a fixed low-pass filter (BlurPool) or a learnable low pass group-wise convolution filter (DDAC).

| | | PASCAL VOC 2012 - DeepLabV3+ (ResNet-18) | | | | | Cityscapes - DeepLabV3+ (ResNet-101) | | | | |
|---|---|---|---|---|---|---|---|---|---|---|---|
| | | Unshifted | Standard Shift | | Circular Shift | | Unshifted | Standard Shift | | Circular Shift | |
| Method | Anti-Alias | mIOU | Consistency | Fidelity | Consistency | Fidelity | mIOU | Consistency | Fidelity | Consistency | Fidelity |
| MaxPool | - | 70.03 | 95.17 | 66.65 | 95.42 | 66.82 | 78.50 | 96.03 | 75.38 | 97.07 | 76.20 |
| Blurpool | LPF-3 | 71.02 | 95.52 | 67.84 | 96.03 | 68.20 | 78.90 | 96.09 | 75.82 | 97.94 | 77.27 |
| DDAC | LPF-3 | 72.28 | 96.77 | 69.95 | 95.98 | 69.37 | 79.52 | 96.28 | 76.54 | 98.21 | 78.09 |
| APS | LPF-3 | 72.37 | 97.05 | 70.24 | 96.70 | 69.98 | 79.84 | 97.53 | 77.87 | 98.32 | 78.50 |
| LPS | LPF-3 | 72.37 | 97.98 | 70.92 | 100.00 | 72.37 | 80.15 | 98.60 | 79.03 | 100.00 | 80.15 |
| **TIPS** | LPF-3 | 73.84 | 98.65 | 72.84 | 100.00 | 73.84 | 81.37 | 99.02 | 80.57 | 100.00 | 81.37 |

Table 4: Semantic segmentation performance on Pascal VOC and Cityscapes datasets.

| | | Kvasir - U-Net | | | | | CVC-ClinicDB - U-Net | | | | |
|---|---|---|---|---|---|---|---|---|---|---|---|
| | | Unshifted | Standard Shift | | Circular Shift | | Unshifted | Standard Shift | | Circular Shift | |
| Method | Anti-Alias | mIOU | Consistency | Fidelity | Consistency | Fidelity | mIOU | Consistency | Fidelity | Consistency | Fidelity |
| MaxPool | - | 75.60 | 92.84 | 70.19 | 97.91 | 74.02 | 73.81 | 90.24 | 66.61 | 95.50 | 70.50 |
| Blurpool | LPF-3 | 78.39 | 94.63 | 74.18 | 98.30 | 77.06 | 76.32 | 93.87 | 71.64 | 96.36 | 73.54 |
| DDAC | LPF-3 | 79.24 | 95.17 | 75.41 | 98.49 | 78.04 | 77.89 | 92.17 | 71.80 | 97.73 | 76.12 |
| APS | LPF-3 | 81.97 | 96.32 | 78.95 | 100.00 | 81.97 | 79.31 | 95.63 | 75.84 | 100.00 | 79.31 |
| LPS | LPF-3 | 82.38 | 97.86 | 80.62 | 100.00 | 82.38 | 78.59 | 96.21 | 75.61 | 100.00 | 78.59 |
| **TIPS** | LPF-3 | 86.10 | 98.09 | 84.46 | 100.00 | 86.10 | 80.05 | 97.89 | 78.36 | 100.00 | 80.05 |

Table 5: Semantic segmentation performance on Kvasir and CVC-ClinicDB datasets.

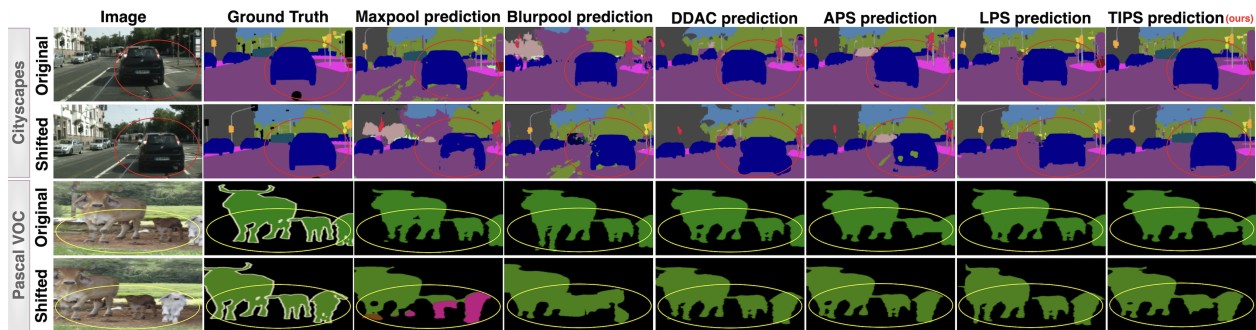

Figure 6: Qualitative comparison of segmentation masks predicted on original and shifted images. Images from Cityscapes, Pascal VOC are standard-shifted by (43,-17), (-38,0) respectively. Regions where TIPS achieve improvements (i.e. consistent segmentation quality) under linear shifts are highlighted with circles.

**Hyperparameters.** We use SGD optimizer with a initial learning rate 0.01, momentum 0.9, weight decay 5e-4 with early stopping. We use DeepLabV3+ (Chen et al., 2018) with ResNet-18 as the backbone for the Pascal-VOC dataset and with ResNet-101 as the backbone for the Cityscapes dataset. For Kvasir and CVC-ClinicDB, we use a UNet (Ronneberger et al., 2015) model with "Kaiming Normal" initialization.

**Evaluation Metrics.** To report shift invariance for semantic segmentation, we use consistency and fidelity similar to image classification experiments, by comparing a common cropped area among images with different shift amounts. Within the common crop, we compute the percentage of pixels that have identical predictions in terms of segmentation categories.

**Results.** Comparison of mIOU, consistency and fidelity in Tables 4, 5 shows that TIPS improves mIOU in comparison to all baselines on all four benchmarks. Consistent with our finding in image classification, we observe a sharper increase in shift consistency under standard shift than with circular. Models trained with TIPS pooling have higher fidelity on both standard and circular shifts, depicting the efficacy of TIPS in learning both shift invariant and high quality segmentation. In Figure 6, we compare the quality of the masks predicted on shifted image when using prior work or TIPS. The areas highlighted with red circles in the first two rows (Cityscapes) demonstrate that TIPS segments objects with higher consistency than other pooling operators under image shifts. The yellow boxes in the last two rows (Pascal-VOC) further illustrate improved segmentation consistency with TIPS under small shifts.

| Stategy | Method | Unshifted Acc. | Standard Shift Consistency | Standard Shift Fidelity | Circular Shift Consistency | Circular Shift Fidelity |
|---|---|---|---|---|---|---|
| *Pooling* | MaxPool | 64.88 | 82.41 | 53.14 | 80.39 | 50.71 |
| *(without anti-aliasing)* | DDAC | 67.59 | 85.43 | 57.74 | 80.90 | 54.68 |
| | APS | 67.05 | 86.39 | 57.92 | **100.00** | 67.05 |
| | LPS | 67.39 | 86.17 | 58.07 | **100.00** | 67.39 |
| | TIPS | **69.02** | **87.42** | **60.34** | **100.00** | **69.02** |
| *Pooling* | BlurPool (LPF-5) | 66.85 | 87.43 | 58.54 | 87.88 | 58.75 |
| *(with LPF-5)* | DDAC (LPF-5) | 66.98 | 86.92 | 58.22 | 80.35 | 53.82 |
| *(anti-aliasing)* | APS (LPF-5) | 67.52 | 87.02 | 58.76 | 99.98 | 67.51 |
| | LPS (LPF-5) | 69.11 | 86.58 | 59.84 | **100.00** | 69.11 |
| | TIPS (LPF-5) | **70.01** | **87.51** | **61.27** | **100.00** | **70.01** |
| *Data Augmentation* | circular | 64.25 | 83.58 | 53.71 | 84.27 | 54.14 |
| | standard | 63.91 | 84.45 | 53.97 | 81.27 | 51.94 |
| | both | **64.87** | **84.99** | **55.13** | **85.64** | **55.55** |
| *Contrastive Learning* | SimCLR | 71.15 | 85.63 | 60.93 | 78.26 | 55.68 |
| | SupCon | **72.49** | **86.17** | **62.46** | **81.75** | **59.26** |

Table 6: A comparison of accuracy and shift consistency and fidelity for additional methods including data augmentation, contrastive learning, and pooling with or without anti-aliasing. The models are trained on the ImageNet dataset with a ResNet18 backbone. Best performance in each section of the table is in bold, performance lower than MaxPool is highlighted in red and overall best performance is in cyan.

## 6 Analysis

We further investigate the effectiveness of TIPS by comparing with non-pooling strategies, conducting ablation studies to examine the impact of our novel loss functions, understanding the effect of hyperparameters, and evaluating the effect of TIPS on various measures of robustness.

### 6.1 Investigating Other Strategies for Improving Shift Invariance of CNNs

In Section 5 we compared TIPS with different pooling methods. There are other approaches besides pooling that could be useful for mitigating failures with pixel-level shift such as data augmentation and contrastive learning. To understand the efficacy of these approaches and compare them with TIPS and other pooling operators, we experiment with three types of data augmentation while training: standard shift, circular shift, and their combination, and two contrastive learning approaches: self-supervised SimCLR (Chen et al., 2020) and supervised SupCon (Khosla et al., 2020). For contrastive learning, representations are learned via the contrastive objective of aligning shifted samples closer and are used for downstream image classification. Table 6 shows a comparison of these techniques with TIPS and previous pooling-based approaches, for image classification with a ResNet-18 backbone, evaluated on the ImageNet dataset. While data augmentation does not significantly improve performance, both contrastive learning methods outperform MaxPool on all metrics. However, TIPS, without any contrastive learning or data augmentation, results in greater shift invariance.

### 6.2 Effect of $\mathcal{L}_{undo}$ and $\mathcal{L}_{FM}$ Regularization

In Figure 7 we analyze the impact of $\mathcal{L}_{undo}$ on learning shift invariant intermediate features, by visualizing $|\psi(X) - X^t|$ at different stages of training. We observe that as training progresses, $\mathcal{L}_{undo}$ is able to guide $|\psi(X) - X^t|$ closer to 0 and thus TIPS learns to offset standard shift transformation on intermediate feature maps. In Figure 8 we quantify this observation by plotting accuracy and four shift invariance measurements for different values of $\epsilon$ in

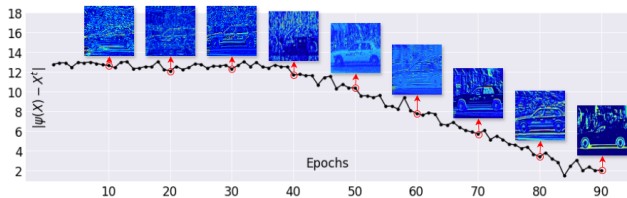

Figure 7: The effect of $\mathcal{L}_{undo}$ in terms of $|\psi(X) - X^t|$ and example feature maps (ResNet-101 with TIPS trained for 90 epochs on ImageNet; $\epsilon$=0.4).

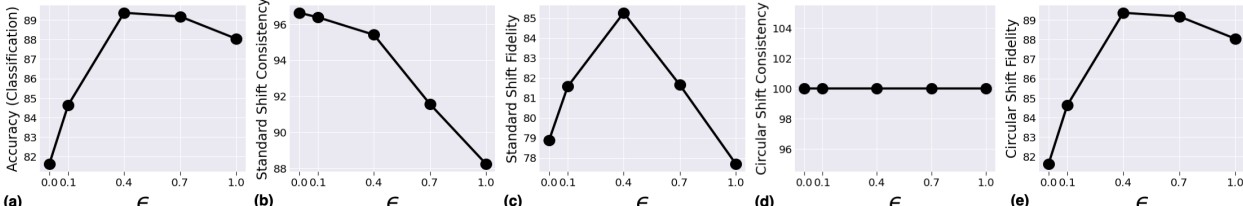

Figure 8: Inspecting the effect of training varying % of epochs on $\mathcal{L}_{undo}$ for Tiny ImageNet classification. Lower $\epsilon$ indicates more epochs with $\mathcal{L}_{undo}$ and vice versa. $\epsilon = 0.4$ (i.e. training without $\mathcal{L}_{undo}$ for the first 40% of epochs and with $\mathcal{L}_{undo}$ for the rest of the epochs is optimal. Values of $\epsilon$ higher than 0.4 yields sub-optimal shift invariance, but is better than low values of $\epsilon$, demonstrating the impact of $\mathcal{L}_{undo}$.

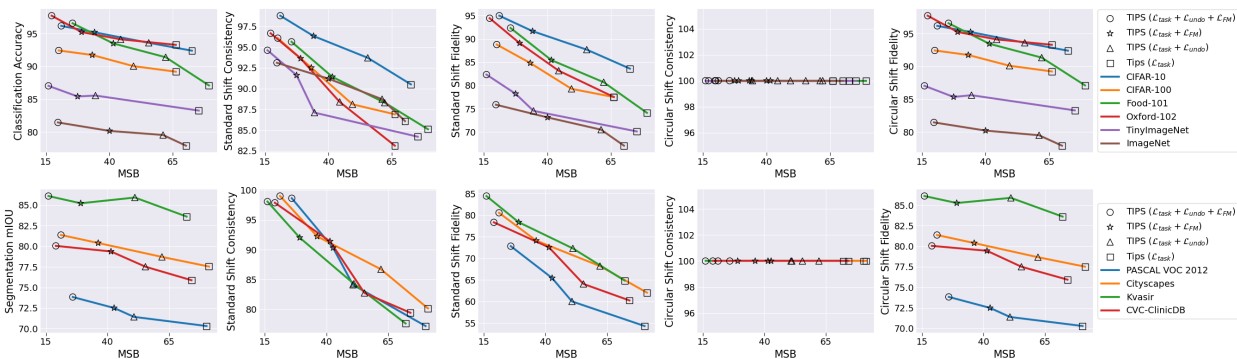

Figure 9: The impact of each component of our training objective is quantified through an ablation study. The top row shows results for six image classification datasets and the bottom row shows results for four semantic segmentation datasets. Regularization using both $\mathcal{L}_{undo}$ and $\mathcal{L}_{FM}$ results in the best performance.

Equation 5. For accuracy and both shift fidelity metrics, $\mathcal{L}_{undo}$ helps, but there is an optimal value of $\epsilon$ (=0.4 for Tiny ImageNet). A very low value of $\epsilon$ hurts performance.

The training objective in Equation 5 includes task loss $\mathcal{L}_{task}$ and two regularizations $\mathcal{L}_{FM}$ and $\mathcal{L}_{undo}$. In Figure 9, we perform an ablation study to examine the efficacy of each term in the loss function. Our results reveal a clear trend: $\circ > \star > \triangle > \square$; across all datasets for image classification and segmentation, regularizing with both $\mathcal{L}_{undo}$ and $\mathcal{L}_{FM}$ (denoted by $\circ$ in the plots) leads to the highest accuracy, highest shift invariance in terms of all four evaluation metrics, and lowest MSB. Only using one of $\mathcal{L}_{undo}$ or $\mathcal{L}_{FM}$ also improves performance compared to training only with $\mathcal{L}_{task}$. These results demonstrate the impact of each component of our loss function on shift invariance and further demonstrate the inverse relationship between MSB and shift invariance.

### 6.3 Effect of the Number of TIPS Layers

Figure 10 (a) portrays mean MSB, consistency, fidelity on Tiny ImageNet classification and Figure 10 (b) shows shift consistency and fidelity on Pascal VOC for semantic segmentation. As we train with more TIPS layer, shift invariance does not always strictly improve, in fact sometimes it decreases. However, for both Tiny ImageNet classification and Pascal VOC semantic segmentation, MSB always decreases as we train with more TIPS layers, indicating the efficacy of using TIPS in reducing MSB.

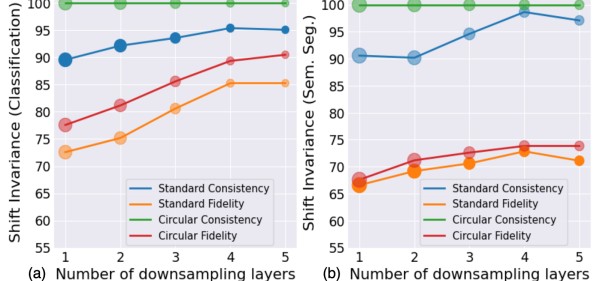

Figure 10: The diameter of the bubbles denotes MSB. As the number of downsampling layers increases, MSB decreases and shift invariance increases.

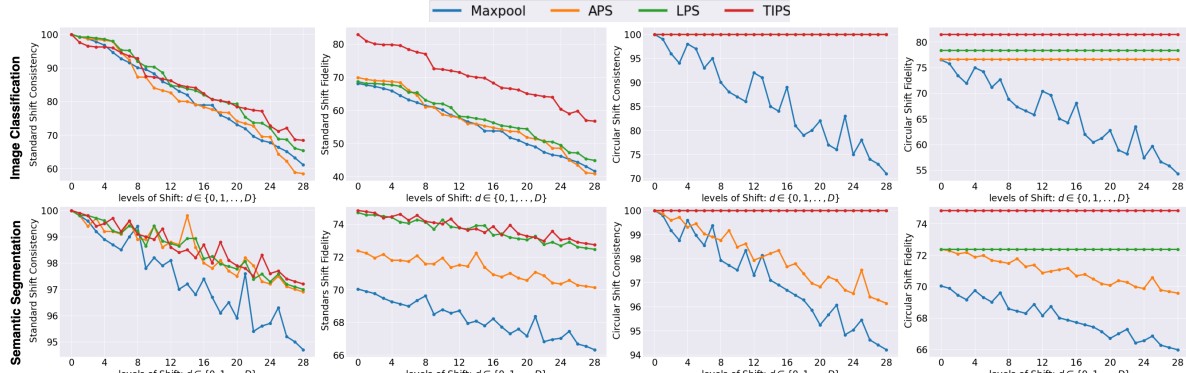

Figure 11: Shift invariance (consistency and fidelity) for standard and circular shifts on image classification (ImageNet) and semantic segmentation (Cityscapes) for varying degrees of shift. TIPS outperform existing pooling operators in all the evaluation metrics for shift invariance.

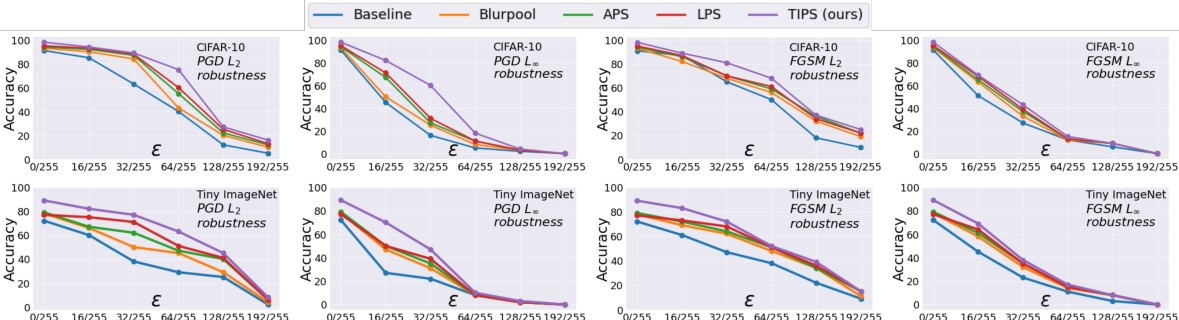

Figure 12: Adversarial robustness under different levels $\varepsilon$ of input perturbations.

### 6.4 Finegrained Results for Different Levels of Shift

In our experiments the level of pixel shift, $d$ is sampled from the range $\{0, 1, ..., D\}$ where $D = h/8$ or $w/8$ for vertical and horizontal shift. In Figure 11, we demonstrate shift invariance under all possible levels of shifts $d \in \{0, 1, ..., D\}$, and observe that shift consistency drops faster with higher degrees of shift when using existing pooling methods whereas with TIPS this degradation is much slower. TIPS not only outperforms other pooling methods on average but at all degrees of shifts $\in \{0, 1, ..., D\}$. We observe that gain with TIPS in comparison to existing pooling operators is higher for shift fidelity than shift consistency. This suggests that TIPS improves both downstream task performance and shift invariance simultaneously.

### 6.5 Robustness Evaluation

**Adversarial Attacks.** Recent studies reveal that deep models with ReLU are vulnerable against adversarial attacks if they are optimized for domain generalization (Frei et al., 2023) or shift invariance (Singla et al., 2021). Studies have also revealed a trade-off between adversarial robustness and other forms of generalization (Gokhale et al., 2022; Moayeri et al., 2022; Teney et al., 2024). We investigate the $\ell_2$ and $\ell_\infty$ adversarial robustness of TIPS (with ResNet-34 backbone trained on CIFAR-10 and Tiny-ImageNet) using PGD (Madry et al., 2018) and FGSM(Goodfellow et al., 2014) attacks from Foolbox (Rauber et al., 2017). Figure 12 shows that TIPS exhibits superior adversarial robustness compared to previous methods. We also observe that better shift invariance is generally correlated with better adversarial robustness

**Patch Attacks.** We adopt the experiment setup from Chaman & Dokmanic (2021) where square patches are randomly erased from the input image and test models trained on the clean CIFAR-10 and ImageNet datasets using a ResNet-18 backbone. Figure 13 demonstrates that TIPS outperforms other methods (pooling and

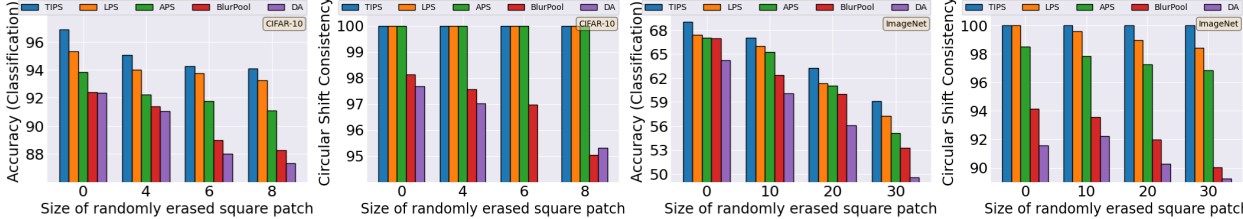

Figure 13: Evaluation of shift invariance under *patch attacks* (randomly erasing image patches) shows that TIPS exhibits higher robustness than existing pooling and data augmentation methods.

| | | | Noise | | | Blur | | | | Weather | | | | Digital | | | |
|---|---|---|---|---|---|---|---|---|---|---|---|---|---|---|---|---|---|
| Method | Clean | mCE | Gauss. | Shot | Impulse | Defocus | Glass | Motion | Zoom | Snow | Frost | Fog | Bright | Contrast | Elastic | Pixel | JPEG |
| VGG-19 | 25.8 | 81.6 | 82.0 | 83.0 | 88.0 | 82.0 | 94.0 | 84.0 | 86.0 | 80.0 | 78.0 | 69.0 | 61.0 | 74.0 | 94.0 | 85.0 | 83.0 |
| VGG-19+**TIPS** | 25.1 | 81.1 | 82.1 | 83.5 | 86.9 | 82.1 | 93.5 | 82.2 | 86.7 | 80.0 | 77.2 | 68.3 | 60.1 | 74.4 | 93.8 | 83.7 | 82.2 |
| ResNet-18 | 30.2 | 84.7 | 87.0 | 88.0 | 91.0 | 84.0 | 91.0 | 87.0 | 89.0 | 86.0 | 84.0 | 78.0 | 69.0 | 78.0 | 90.0 | 80.0 | 85.0 |
| ResNet-18+**TIPS** | 28.7 | 83.9 | 85.3 | 87.9 | 91.6 | 83.6 | 91.2 | 85.7 | 88.3 | 85.4 | 82.6 | 77.1 | 68.5 | 77.3 | 89.9 | 80.1 | 84.6 |

Table 7: Errors on clean (ImageNet) and corrupted (ImageNet-C) test sets. mCE is the mean corruption error. Models are trained only on clean ImageNet training dataset.

data augmentation) in robustness to such patch attacks. On ImageNet, shift consistency is more pronounced than other methods, especially for larger erased patches.

**Natural Corruptions.** We evaluated the robustness of TIPS under an *out-of-distribution* setting, where models are trained on clean images, but tested on images with natural corruptions due to noise, blur, weather artifacts, or digital corruptions. We test robustness to natural corruptions using the ImageNet-C test dataset (Hendrycks & Dietterich, 2018) and report **error** on clean ImageNet (complement of classification accuracy). Table 7 shows that with TIPS, the mCE (mean corruption error) for VGG-19 and ResNet-18 architectures decreased by 0.61 % and 0.94 % respectively.

## 6.6 Applicability of TIPS to Vision Transformers

Our work is focused on improving shift invariance of CNNs – models that are already in use in may real-world applications. We note that in vision tranformers, three modules break shift invariance:

- *Patch embeddings* convert image patches into vectors using strided convolution (not shift invariant).
- *Positional encodings* for both shifted and non-shifted inputs are identical (amount of shift is not encoded).
- *Window-based self-attention* is computationally cheap, but applying local attention on windows of sizes larger than amount of input shift causes token values to change invariantly w.r.t. input shift.

Since these mechanisms are not analogous to downsampling, polyphase sampling cannot be directly applied to ViTs as conveniently as CNNs. Although TIPS is currently limited to CNNs, in our experiments we show that ViTs are also not shift invariant and our simple plug-in solution for CNNs (TIPS) outperforms ViTs.

## 7 Conclusion

Through a large scale correlation analysis we identify a strong inverse relationship of shift invariance of convolutional neural networks with the maximum-sampling bias (MSB) of pooling operators. We find that optimizing neural network weights to reduce MSB is a good strategy for improving shift invariance. With our proposed learnable Translation Invariant Polyphase Sampling (TIPS) pooling layer and regularization that promotes low MSB, we achieve state-of-the-art results for shift invariance on a variety of image classification and semantic segmentation benchmarks, outperforming data augmentation and contrastive learning strategies. Our analysis reveals additional benefits of TIPS, including improved robustness to adversarial attacks and corruptions. Our work serves as a starting point for further empirical or theoretical investigations into factors that cause sensitivity to shift.

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

## A    Appendix

In this appendix, we define standard and circular shifts of images with examples. We further discuss computational analysis and experimental setup for MSB - shift invariance correlation study, image classification benchmarks, semantic segmentation benchmarks. Finally, we illustrate the computational overhead in TIPS and discuss how it compares to existing pooling operators.

### A.1   Standard and Circular Shifts of Images

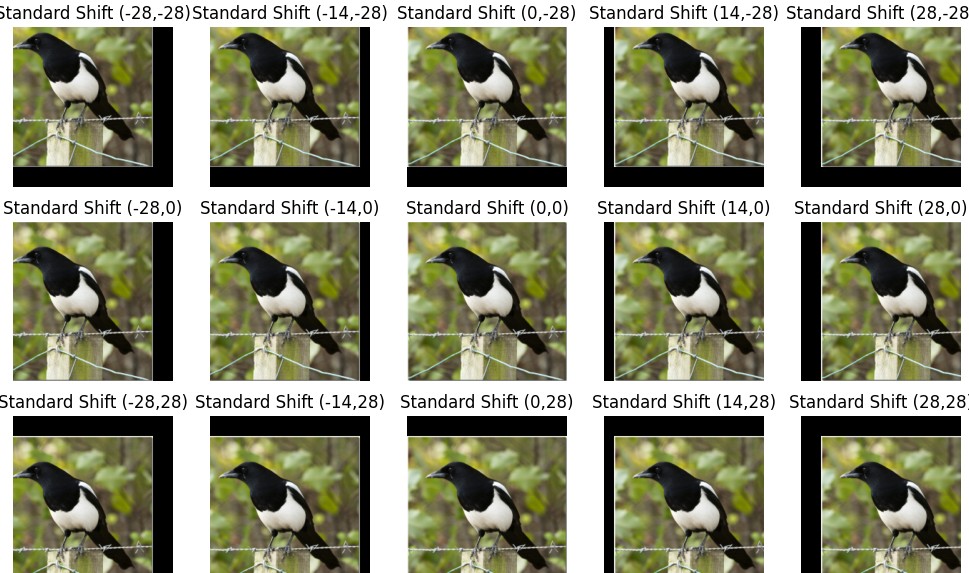

Figure 14: Standard shift of an $224 \times 224$ image from ImageNet test set is shown with varying amount of shifts. Here, standard shift $(0,0)$ denotes the original image with no shifts. It is also observed that, as the amount of standard shift increases, there occurs more information (pixel) loss.

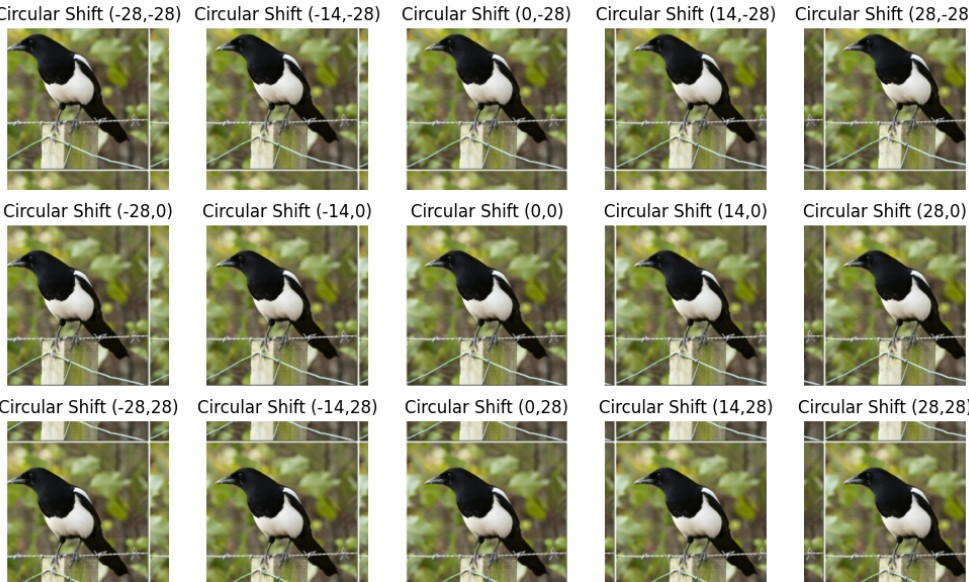

Figure 15: Circular shift of an $224 \times 224$ image from ImageNet test set is shown with varying amount of shifts. Here, circular shift $(0, 0)$ denotes the original image with no shifts.

There are two types of pixel levels shifts that can be performed on images: standard shift and circular shift. Given an image of height $h$ and width $w$, we can perform either type of shifts by an amount $(x, y)$ where $x \in \{0, .., h\}$, $y \in \{0, .., w\}$. Standard shift is the process of shifting images to a $(x, y)$ direction which renders blank pixels at shifted positions. Circular shift also shifts images in the $(x, y)$ direction, except the shifted pixels that move beyond the image boundary, are wrapped about the opposite ends of the image to fill in the empty pixels. Therefore, circular shift is a lossless transformation while standard shift is not. Figure 14 and 15 show examples of standard and circular shift (by varying amounts) applied to an image taken from ImageNet test set and depict how standard shift renders blank pixels while circular shift do not.

## A.2 Experimental Setup for MSB - Shift Invariance Correlation Study

Table 8 shows the list of CNN architectures (including Mobile Net by Howard et al. (2017)), datasets and pooling methods that we use to obtain a total of 576 models for the MSB-shift invariance correlation study. In our study, we train each combination of architecture and dataset on 9 pooling methods: Global Average Pooling before classification with no spatial downsampling of convolutional features, TIPS ($\epsilon = 0.4, \alpha = 0.35$ for image classification, $\epsilon = 0.45, \alpha = 0.35$ for semantic segmentation), LPS ($\tau = 0.01$), APS ($p = 2$), APS ($p \to \infty$), LPS ($\tau \to \infty$), BlurPool (LPF-5), Average Pool ($2 \times 2$), and MaxPool ($2 \times 2$). Furthermore, in each of the aforementioned settings, we use different number of pooling layers as shown in Table 8. While training with Global Average Pooling, we use 4 different kernel sizes ($2 \times 2, 3 \times 3, 4 \times 4, 5 \times 5$) in the first convolution layer with *same padding* to create 4 variants since varying the number pooling layers is not possible in this setting barring that we downsample only once (downsampling the very last convolution features with Global Average Pooling before classification/segmentation layer).

In Table 9, Table 10 we include training details such as image size, batch size, step size, number of training epochs for all model - dataset combinations used in the MSB - shift invariance correlation framework for both image classification and semantic segmentation. As discussed in Section 4, using Global Average Pooling with no spatial downsampling of the convolution features leads to increased computation with larger spatial features. In Table 11, we summarize a detailed analysis on how Global Average Pooling increases computational complexity in comparison to baseline MaxPool. The reported CUDA time is in *nanoseconds (ns)*, CUDA memory is in *Mega Bytes (MB)*, GFLOPs is *billions* of floating point operations per second. In Figure 5, we observe that Global Average Pooling improves shift invariance and reduces MSB, but Table 11

| Image Classification Experiments | | | Semantic Segmentation Experiments | | |
|---|---|---|---|---|---|
| **Model** | **# Layers** | **Dataset** | **Model** | **# Layers** | **Dataset** |
| MobileNet | $\{2, 3, 4, 5\}$ | CIFAR-10 | DeepLabV3+ (ResNet-18) | $\{2, 3, 4, 5\}$ | PASCAL VOC 2012 |
| ResNet-18 | $\{2, 3, 4, 5\}$ | CIFAR-100 | DeepLabV3+ (ResNet-101) | $\{3, 4, 5.6\}$ | Cityscapes |
| ResNet-34 | $\{2, 3, 4, 5\}$ | Food-101 | U-Net (ResNet-18) | $\{2, 3, 4, 5\}$ | Kvasir |
| ResNet-101 | $\{2, 3, 4, 5\}$ | Oxford-102 | U-Net (ResNet-34) | $\{2, 3, 4, 5\}$ | CVC-ClinicDB |

Table 8: List of CNN architectures and datasets, tested on each pooling method for correlation analysis between MSB and Shift Invariance.

| | CIFAR-10 | | | | CIFAR-100 | | | | Food-101 | | | | Oxford-102 | | | |
|---|---|---|---|---|---|---|---|---|---|---|---|---|---|---|---|---|
| **Model** | $h \times w$ | $b$ | $s$ | $N$ | $h \times w$ | $b$ | $s$ | $N$ | $h \times w$ | $b$ | $s$ | $N$ | $h \times w$ | $b$ | $s$ | $N$ |
| MobileNet | 32×32 | 64 | 60 | 220 | 32×32 | 64 | 60 | 220 | 200×200 | 128 | 60 | 220 | 200×200 | 128 | 60 | 220 |
| ResNet-18 | 32×32 | 64 | 50 | 250 | 32×32 | 64 | 50 | 250 | 224×224 | 64 | 50 | 250 | 224×224 | 64 | 50 | 250 |
| ResNet-34 | 32×32 | 64 | 50 | 250 | 32×32 | 64 | 50 | 250 | 224×224 | 64 | 50 | 250 | 224×224 | 64 | 50 | 250 |
| ResNet-101 | 32×32 | 64 | 180 | 480 | 32×32 | 64 | 180 | 480 | 224×224 | 64 | 180 | 480 | 224×224 | 64 | 180 | 480 |

Table 9: Image size ($h \times w$), batch size ($b$), step size($s$) for updating learning rate, and number of epochs ($N$) reported for each CNN model and image classification dataset combination for the MSB – Shift Invariance correlation analysis experiment.

| | Pascal VOC 2012 | | | | Cityscapes | | | | Kvasir | | | | CVC-ClinicDB | | | |
|---|---|---|---|---|---|---|---|---|---|---|---|---|---|---|---|---|
| **Model** | $h \times w$ | $b$ | $s$ | $N$ | $h \times w$ | $b$ | $s$ | $N$ | $h \times w$ | $b$ | $s$ | $N$ | $h \times w$ | $b$ | $s$ | $N$ |
| DeepLabV3+ (ResNet-18) | 200×300 | 12 | 120 | 450 | 200×200 | 12 | 120 | 450 | 200×200 | 12 | 60 | 450 | 200×300 | 8 | 45 | 450 |
| DeepLabV3+ (ResNet-101) | 200×300 | 8 | 120 | 380 | 200×200 | 12 | 120 | 380 | 200×200 | 12 | 60 | 380 | 200×300 | 8 | 45 | 380 |
| U-Net (ResNet-18) | 200×300 | 12 | 120 | 180 | 200×200 | 16 | 120 | 180 | 200×200 | 16 | 60 | 180 | 200×300 | 12 | 45 | 180 |
| U-Net (ResNet-34) | 200×300 | 12 | 120 | 150 | 200×200 | 12 | 120 | 150 | 200×200 | 12 | 60 | 150 | 200×300 | 8 | 45 | 150 |

Table 10: Image size ($h \times w$), batch size ($b$), step size($s$) for updating learning rate, and number of epochs ($N$) reported for each CNN model and semantic segmentation dataset combination for the MSB – Shift Invariance correlation analysis experiment.

reveals that this performance gain comes at a significantly higher computational cost. However, with TIPS we achieve comparable shift invariance and MSB by introducing marginal computational complexity in comparison to Global Average Pooling.

## A.3 Experimental Setup for Image Classification and Semantic Segmentation

We benchmark the performance of TIPS and prior work on six image classification datasets which are described in Table 12. We benchmark the performance of TIPS and prior work on four semantic segmentation datasets which are described in Table 13. Table 12, 13 contains further training details on all the reported datasets such as batch size, step size, number of training epochs, image/crop size, number of classes and number of images in the dataset.

## A.4 Computational Overhead in TIPS

Table 14 shows the percentage of additional parameters required to use TIPS on image classification and segmentation CNN models with different pooling methods and CNN architectures, for RGB images of size $224 \times 224$ and a batch-size of 64. TIPS introduces marginal computational overhead while still being computationally cheaper than existing pooling operators for shift invariance, i.e. DDAC. Moreover, in Table 15 we show the number of trainable parameters with different pooling operators for all the image classification, semantic segmentation CNN models. While TIPS requires higher number of trainable parameters than LPS, it is still much less than DDAC.

| Architecture | Pooling | CUDA Time ↓ | CUDA Memory ↓ | GFLOPs ↓ | Architecture | Pooling | CUDA Time ↓ | CUDA Memory ↓ | GFLOPs ↓ |
|---|---|---|---|---|---|---|---|---|---|
| MobileNet | MaxPool | 0.635 | 58.122 | 2.270 | DeepLabV3+(ResNet-18) | MaxPool | 1.33 | 25.216 | 9.570 |
| | TIPS | 1.045 | 101.214 | 3.005 | | TIPS | 3.728 | 380.146 | 91.146 |
| | GAP | 66.609 | 6390.284 | 639.259 | | GAP | 121.029 | 2453.834 | 1926.592 |
| ResNet-18 | MaxPool | 1.135 | 21.860 | 4.017 | DeepLabV3+(ResNet-101) | MaxPool | 7.52 | 144.737 | 51.447 |
| | TIPS | 3.525 | 292.844 | 41.937 | | TIPS | 18.274 | 911.845 | 246.947 |
| | GAP | 72.460 | 1957.691 | 1124.032 | | GAP | 741.568 | 21671.086 | 11521.953 |
| ResNet-34 | MaxPool | 1.954 | 31.904 | 8.128 | U-Net(ResNet-18) | MaxPool | 2.754 | 78.574 | 23.567 |
| | TIPS | 5.623 | 334.754 | 71.532 | | TIPS | 8.675 | 1113.227 | 235.797 |
| | GAP | 141.075 | 3451.912 | 2250.287 | | GAP | 143.402 | 3137.765 | 2700.095 |
| ResNet-101 | MaxPool | 4.921 | 131.035 | 31.197 | U-Net(ResNet-34) | MaxPool | 3.179 | 88.707 | 27.678 |
| | TIPS | 12.204 | 816.791 | 146.596 | | TIPS | 9.045 | 957.965 | 246.352 |
| | GAP | 534.514 | 21144.011 | 8508.809 | | GAP | 202.312 | 4631.986 | 3826.350 |

(a) Image Classification | (b) Semantic Segmentation

Table 11: GPU resources (CUDA time, memory, GFLOPs) allocated to convolution operations in CNNs while using different pooling operators for various CNN architectures. We observe that, performing Global Average Pooling (GAP) on the final convolution feature with no prior downsampling drastically increases GPU resources in comparison to baseline MaxPool. TIPS require additional convolution layers (Figure 3), since it is a learnable pooling operator. Compared to MaxPool, the overhead in GPU resources with TIPS is remarkably smaller than it is for Global Average Pooling.

| | | **Image Classification Experiments** | | | | | | |
|---|---|---|---|---|---|---|---|---|
| Dataset | Model | Batch Size | Step Size | Epochs | Image Size | # Classes | # Training Samples | # Validation Samples |
| CIFAR-10 | ResNet-18 | 64 | 50 | 250 | $32\times32$ | 10 | 50,000 | 10,000 |
| CIFAR-100 | ResNet-34 | 64 | 50 | 250 | $32\times32$ | 100 | 50,000 | 10,000 |
| Food-101 | ResNet-50 | 64 | 25 | 80 | $224\times224$ | 101 | 75,750 | 25,250 |
| Oxford-102 | ResNet-50 | 64 | 20 | 70 | $224\times224$ | 102 | 2,060 | 6,129 |
| Tiny ImageNet | ResNet-101 | 64 | 180 | 480 | $64\times64$ | 200 | 100,000 | 10,000 |
| ImageNet | ResNet-101 | 64 | 30 | 90 | $224\times224$ | 1000 | 1,281,167 | 50,000 |

Table 12: Training details, dataset statistics for all six datasets in our image classification experiments. Training details include batch size, step size for updating learning rate, number of training epochs, image size and dataset statistics include number of classes, training samples, validation samples.

| | | **Semantic Segmentation Experiments** | | | | | | |
|---|---|---|---|---|---|---|---|---|
| Dataset | Model | Batch Size | Step Size | Epochs | Image Size | # Classes | # Training Samples | # Validation Samples |
| PASCAL VOC 2012 | DeepLabV3+(ResNet-18) | 12 | 120 | 450 | $200\times300$ | 20 | 1,464 | 1,456 |
| Cityscapes | DeepLabV3+(ResNet-101) | 12 | 120 | 380 | $200\times200$ | 19 | 2,975 | 500 |
| Kvasir | UNet(ResNet-18) | 12 | 60 | 180 | $200\times200$ | 2 | 850 | 150 |
| CVC-ClinicDB | UNet(ResNet-34) | 8 | 45 | 150 | $200\times300$ | 2 | 521 | 91 |

Table 13: Training details, dataset statistics for all four datasets in our semantic segmentation experiments. Training details include batch size, step size for updating learning rate, number of training epochs, image size and dataset statistics include number of classes, training samples, validation samples.

## A.5 Effect of training on $\mathcal{L}_{FM}$

In Figure 16, we train ResNet-101 on Tiny ImageNet with TIPS and $\mathcal{L}_{FM}$ and compare it with baselines LPS, APS and MaxPool in terms of standard fidelity and MSB. To further inspect the effect of training TIPS with $\mathcal{L}_{FM}$, we train with three different setting of TIPS: (1) TIPS with $\mathcal{L}_{FM}$: to discourages both skewed and uniform $\tau$, (2) TIPS with only the first term in $\mathcal{L}_{FM}$: to discourages skewed $\tau$ only, and (3) TIPS with only second term in $\mathcal{L}_{FM}$: to discourages uniform $\tau$ only. We observe that training TIPS with both terms from $\mathcal{L}_{FM}$ yields the maximum gain in shift fidelity and decreases MSB the most. TIPS with $\mathcal{L}_{FM}$ also outperforms other pooling methods: LPS, APS and MaxPool in terms of standard shift fidelity and MSB.

| Method | ResNet-18 | ResNet-34 | ResNet-50 | ResNet-101 |
|---|---|---|---|---|
| BlurPool | 0.00 | 0.00 | 0.00 | 0.00 |
| DDAC | 7.92 | 10.53 | 9.27 | 4.30 |
| APS | 0.00 | 0.00 | 0.00 | 0.00 |
| LPS | 1.03 | 2.24 | 1.93 | 1.05 |
| **TIPS** | 5.51 | 4.56 | 2.17 | 3.19 |

(a) Image Classification

| Method | DeepLabV3+(A) | DeepLabV3+(B) | UNet |
|---|---|---|---|
| BlurPool | 0.00 | 0.00 | 0.00 |
| DDAC | 12.00 | 4.83 | 12.83 |
| APS | 0.00 | 0.00 | 0.00 |
| LPS | 4.40 | 3.25 | 4.79 |
| **TIPS** | 7.24 | 4.04 | 5.76 |

(b) Semantic Segmentation

Table 14: Percentage of additional parameters required in comparison to MaxPool on each CNN architecture for classification and semantic segmentation. We observe that, while TIPS require more parameters than LPS, DDAC causes the maximum increase in trainable parameters *w.r.t.* baseline MaxPool.

| Method | ResNet-18 | ResNet-34 | ResNet-50 | ResNet-101 |
|---|---|---|---|---|
| MaxPool | 11.884 | 21.282 | 23.521 | 42.520 |
| BlurPool | 11.884 | 21.282 | 23.521 | 42.520 |
| DDAC | 12.825 | 23.524 | 25.701 | 44.349 |
| APS | 11.884 | 21.282 | 23.521 | 42.520 |
| LPS | 12.006 | 21.759 | 23.975 | 42.966 |
| **TIPS** | 12.539 | 22.253 | 24.031 | 43.876 |

(a) Image Classification

| Method | DeepLabV3+(A) | DeepLabV3+(B) | UNet |
|---|---|---|---|
| MaxPool | 20.131 | 58.630 | 7.762 |
| BlurPool | 20.131 | 58.630 | 7.762 |
| DDAC | 22.547 | 61.459 | 8.758 |
| APS | 20.131 | 58.630 | 7.762 |
| LPS | 21.017 | 60.536 | 8.134 |
| **TIPS** | 21.589 | 60.999 | 8.209 |

(b) Semantic Segmentation

Table 15: Number of trainable parameters in **Million** for various pooling methods reported for: ResNet-18, ResNet-34, ResNet-50, ResNet-101 backbones (image classification), DeepLabV3+ (A: ResNet-18, B: ResNet-101) and UNet (semantic segmentation). Number of trainable parameters are computed assuming an RGB input image of size $224 \times 224$.

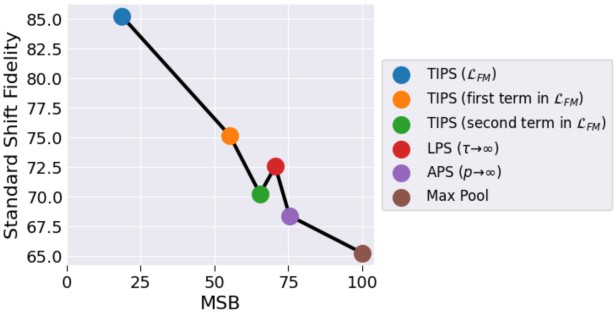

Figure 16: The effect of $\mathcal{L}_{FM}$ on TIPS is visualized by plotting standard shift fidelity versus MSB for models trained on Tiny ImageNet. Training TIPS with $\mathcal{L}_{FM}$ yields the maximum standard shift fidelity and minimum MSB.

