# OpenReview forum: "Improving Shift Invariance in Convolutional Neural Networks with Translation Invariant Polyphase Sampling"
_TMLR — Rejected by TMLR_

### Review · Reviewer_gkKW · 2024-05-21

**Summary Of Contributions:**

This paper introduces a metric called Maximum-Sampling Bias (MSB), and shows that it has a strong negative correlation with shift-invariance. Moreover, the paper proposes the Translation Invariant Polyphase Sampling (TIPS) pooling layer to reduce MSB (improves shift invariance) under shift transformations, L_FM loss function to decrease failure modes of shift invariance, and L_undo loss function for undoing standard shifts. Finally, it demonstrates that the proposed approach outperforms the state-of-the-art pooling operations under shift transformation in the image classification and semantic segmentation tasks.

**Audience:**

Yes

**Claims And Evidence:**

No

**Requested Changes:**

Please see the weaknesses!

**Strengths And Weaknesses:**

**Strengths**:
Well-written and well-presented

**Weaknesses**:

**Major**:

1. The need for a new shift-invariance related metric such as MSB seems unclear to me. There are already other metrics for shift-invariance (e.g. shift consistency and fidelity), which have a strong negative correlation with MSB as the authors mentioned. Why shouldn't we use them in our analysis directly instead of using MSB. I mean instead of saying lower MSB improves the model performance, we can say higher shift consistency/fidelity leads to higher model performance.

2. The accuracy results for the CIFAR-100 dataset seem too high. Based on previous studies [1, 2], ResNets can achieve maximum accuracy of around 80% on CIFAR-100. In Table-1, the authors reported accuracy of 88.38% for max-pooling based ResNet-34. Moreover, I am not aware of any study which reports accuracy results using ResNet-34 on CIFAR-100, consistent with those reported in this paper. The authors should provide a detailed description of their training procedure, or mention the baseline study they followed for training the models if that is the case.


3. According to Figure 5, global average pooling gives the lowest MSB and highest accuracy compared to the other pooling layers including TIPS. Why should we use TIPS instead of global average pooling?

4. The proposed method requires much more trainable parameters than many of the state-of-the-art pooling methods. For instance, TIPS based ResNet-34 has around 1 million more parameters than Max-pooling based counterpart according to Table-15. How can we ensure that improvement from TIPS is not because of more trainable parameters?

5. For image classification using CNNs, only ResNet architecture has been used in the experiments. More experiments are required using different architectures such as DenseNets or EfficientNets.

6. The normalization layer highly impacts the performance of CNNs. The authors only used the default BatchNorm [3] layer for CNNs in their experiments. It would be great if the authors provide some initial results using other normalization layers such as GroupNorm [4] or recently proposed KernelNorm [2].

7. The difference between the previous approaches such as APS and LPS and the proposed approach of TIPS is unclear. The author should elaborate more on that.

**Minor**:
1. Figure 2 and the corresponding description are confusing: It seems that x and x' are identical inputs, i.e. transformation g is an identity mapping. However, they need to be different in order to see how the model output changes. I think the statement in the paper is enough to get the general idea behind shift-invariance and shift-equivariance: "Shift-invariance is desirable for image classification to ensure that categorical outputs are invariant to pixel shift, and shift-equivariance is desirable for semantic segmentation to ensure that pixel-shift in the image results in equivalent shift in segmentation".

2. Section 3: "A pooling layer with stride s, downsamples X into ...". It should be specified that kernel size and stride values are identical and padding is zero.

3. Last sentence of page 4: "using the using Kaiming normal approach" -> "using the Kaiming normal approach"

4. The authors chose ϵ = 0.4 and α = 0.35 in the experiments. They should clarify how those values for ϵ and α have been obtained. Using parameter tuning?

1) Huang et al., "Densely connected convolutional networks", CVPR, 2017, https://arxiv.org/pdf/1608.06993
2) Nasirigerdeh et al., "Kernel Normalized Convolutional Networks", TMLR, 2024, https://openreview.net/pdf?id=Uv3XVAEgG6
3) Ioffe et al., "Batch normalization: Accelerating deep network training by reducing internal covariate shift", ICML, 2015, http://proceedings.mlr.press/v37/ioffe15.pdf
4) Wu et al., "Group Normalization", ECCV, 2018, https://openaccess.thecvf.com/content_ECCV_2018/papers/Yuxin_Wu_Group_Normalization_ECCV_2018_paper.pdf

---

> ### Author Response · Authors · 2024-06-17
> **Response #1 to reviewer gkKW: Major Issue 1, 2, 3**
>
> **Major Issue 1:** The need for a new shift-invariance related metric such as MSB seems unclear to me. There are already other metrics for shift-invariance (e.g. shift consistency and fidelity), which have a strong negative correlation with MSB as the authors mentioned. Why shouldn't we use them in our analysis directly instead of using MSB.
> **Response:**
>
> We identify maximum-sampling bias (MSB) as a factor that affects downstream performance in metrics such as shift consistency and shift fidelity.  Note that MSB is a property of existing pooling operators and does not require evaluation of models on datasets. It is true that MSB is negatively correlated with shift consistency/fidelity – this is in fact a key finding from our work. This means that MSB (which does not require empirical evaluation on datasets) is merely a property of the pooling operators and can be used to reason about shift invariance of a convolutional neural network architecture.  This finding entails that if upper and lower bounds on MSB can be quantified, future work will be able to potentially derive theoretical guarantees on shift consistency and fidelity without requiring empirical results on datasets. Our hope is that a concept such as MSB will pave the way towards such theoretical results.
>
> **Major Issue 2:** The accuracy results for the CIFAR-100 dataset seem too high. Based on previous studies, ResNets can achieve maximum accuracy of around 80% on CIFAR-100. The authors should provide a detailed description of their training procedure, or mention the baseline study they followed for training the models if that is the case.
> **Response:**
>
> To ensure fair comparison with prior work, we train each CNN model using identical optimization settings and hyperparameters, as mentioned in Sec 5.1 and Table 12 (Appendix).
> Our TMLR submission is accompanied by code in the supplementary materials. We will make our code, model weights, and results public after the anonymity period ends.
>
> **Major Issue 3:** According to Figure 5, global average pooling gives the lowest MSB and highest accuracy compared to the other pooling layers including TIPS. Why should we use TIPS instead of global average pooling?
> **Response:**
>
> As stated in the paragraph “Negative Correlation between MSB and Shift Invariance” on page 6, the cases where least MSB and highest performances are observed uses Global Average Pooling (GAP) with no spatial pooling in the intermediate layers. However, using only GAP with no intermediate pooling due to having larger grids to convolve increases CUDA time (106 times), memory (110 times) and GFLOPS (281 times) in comparison to baseline MaxPool (shown in Tab 11 for MobileNet-classification). Whereas TIPS increases CUDA time (1.64 times), memory (1.74 times) and GFLOPS (1.32 times) when used instead of baseline MaxPool (shown in Tab 11 for MobileNet-classification). We show a detailed comparison among these three choices for pooling (maxpool, TIPS, GAP with no intermediate pooling) in Tab 11 which reveals that GAP exhibits additional computation in comparison to TIPS across the board on a number of image classification and semantic segmentation tasks with various CNN architectures.
> Tab 11 demonstrates that TIPS is a compute efficient choice to improve shift consistency and fidelity in comparison to using GAP with no intermediate pooling.

---

> ### Author Response · Authors · 2024-06-17
> **Response #2 to reviewer gkKW: Major Issue 4**
>
> **Major Issue 4:** The proposed method requires much more trainable parameters than many of the state-of-the-art pooling methods. For instance, TIPS based ResNet-34 has around 1 million more parameters than Max-pooling based counterpart according to Table-15. How can we ensure that improvement from TIPS is not because of more trainable parameters?
> **Response:**
>
> Table A shows a comparison between MaxPool and TIPS using CNN architectures with different number of parameters.  We observe that with MaxPool as the pooling operator, even a >400% increase in parameters (from 11.9M in ResNet18 to 64.0M in EfficientNet) only results in a 1.71% improvement in shift consistency.  Meanwhile, using TIPS with ResNet18 (a 5% increase in number of parameters from 11.9M to 12.5M), leads to a 11.22% increase in shift consistency.  This is observed across the board for different architectures and datasets.
>
> | Pooling         | MaxPool                   |                          | TIPS                     |                          |
> |-----------------|---------------------------|--------------------------|--------------------------|--------------------------|
> | **CNN Model**       | **Parameters (M)**            | **Standard Shift Consistency** | **Parameters (M)**            | **Standard Shift Consistency** |
> | ResNet-18       | 11.9                      | 87.43                    | 12.5                     | 98.65                    |
> | DenseNet-BC (k=24) [1] | 15.3                      | 90.02                    | 16.9                     | 96.71                    |
> | ResNet-34       | 21.3                      | 88.93                    | 22.3                     | 98.60                    |
> | EfficientNet-B7 [2] | 64.0                      | 89.14                    | 67.3                     | 93.67                    |
> ||
> **Table A:** Inspecting how standard shift consistency in CIFAR-10 improves by increasing trainable parameters (bigger CNN models and switching pooling layers from MaxPool to TIPS).

---

> ### Author Response · Authors · 2024-06-17
> **Response #3 to reviewer gkKW: Major Issue  5**
>
> **Major Issue 5:** On experimenting CNNs beyond ResNets, i.e. DenseNets or EfficientNets.
> **Response:**
>
> Tab B, C, D, E contain results from DenseNet [1] and EfficientNet [2] on CIFAR-10 and CIFAR-100 with different pooling methods including TIPS.
> We use LPF-5 for antialiasing and hyperparameters used in DenseNet and EfficientNet respectively.
> We observe improved shift invariance with TIPS independent of CNN architecture.
>
> | **DenseNet-BC (k=24) [1]** | CIFAR-10        | Standard |  Shift                 | Circular |   Shift                |
> |----------------------------|-----------------|----------------|-------------------|----------------|-------------------|
> | **Pooling Method**         | **Accuracy**    | **Consistency**     | **Fidelity**      | **Consistency**     | **Fidelity**      |
> | MaxPool                    | 96.37           | 90.02          | 86.72             | 92.41          | 89.05             |
> | BlurPool                   | 96.74           | 92.57          | 89.51             | 94.07          | 90.96             |
> | APS                        | 97.27           | 93.31          | 90.82             | **100.00**         | 97.27             |
> | LPS                        | 97.12           | 94.36          | 91.62             | **100.00**         | 97.12             |
> | TIPS (ours)                | **97.43**       | **96.71**      | **94.19**         | **100.00**     | **97.43**         |
> ||
> **Table B:**  Image classification performance on CIFAR-10 with DenseNet-BC (k=24).
>
> | **DenseNet-BC (k=24) [1]** | CIFAR-100        | Standard |  Shift                 | Circular |   Shift                |
> |----------------------------|-----------------|---------------------|-------------------|---------------------|-------------------|
> | **Pooling Method**         | **Accuracy**    | **Consistency**     | **Fidelity**      | **Consistency**     | **Fidelity**      |
> | MaxPool                    | 80.35           | 92.72               | 74.58             | 89.63               | 72.02             |
> | BlurPool                   | 81.92           | 93.41               | 76.54             | 90.86               | 74.47             |
> | APS                        | 83.29           | 94.27               | 78.52             | **100.00**              | 83.29             |
> | LPS                        | 83.87           | 95.31               | 79.91             | **100.00**              | 83.87             |
> | TIPS (ours)                | **83.89**       | **96.17**           | **80.64**         | **100.00**          | **83.89**         |
> ||
> **Table C:**  Image classification performance on CIFAR-100 with DenseNet-BC (k=24).
>
> | **EfficientNet-B7 [2]**          | CIFAR-10        | Standard |  Shift                 | Circular |   Shift                |
> |----------------------------|-----------------|---------------------|-------------------|---------------------|-------------------|
> | **Pooling Method**         | **Accuracy**    | **Consistency**     | **Fidelity**      | **Consistency**     | **Fidelity**      |
> | **MaxPool**                      | 98.90           | 89.14               | 88.13             | 92.19               | 91.22             |
> | **BlurPool**                     | 98.90           | 91.06               | 90.07             | 92.37               | 91.39             |
> | **APS**                          | 98.53           | 92.30               | 90.95             | **100.00**              | 98.53             |
> | **LPS**                          | **98.93**       | 93.47               | 92.44             | **100.00**              | **98.93**         |
> | **TIPS (ours)**                  | **98.93**       | **93.67**           | **92.67**         | **100.00**          | **98.93**         |
> ||
> **Table D:**  Image classification performance on CIFAR-10 with EfficientNet-B7.
>
> | **EfficientNet-B7 [2]**          | CIFAR-100       | Standard |  Shift                 | Circular |   Shift                |
> |----------------------------|-----------------|---------------------|-------------------|---------------------|-------------------|
> | **Pooling Method**         | **Accuracy**    | **Consistency**     | **Fidelity**      | **Consistency**     | **Fidelity**      |
> | MaxPool                    | 91.72           | 93.51               | 85.76             | 90.25               | 82.78             |
> | BlurPool                   | 92.17           | 94.68               | 87.27             | 92.39               | 85.20             |
> | APS                        | 92.18           | 94.57               | 87.31             | **100.00**              | 92.18             |
> | LPS                        | 93.50           | 95.81               | 89.60             | **100.00**              | 93.50             |
> | TIPS (ours)                | **93.71**       | **96.04**           | **89.99**         | **100.00**          | **93.71**         |
> ||
> **Table E:**  Image classification performance on CIFAR-100 with EfficientNet-B7.

---

> ### Author Response · Authors · 2024-06-17
> **Response #4 to reviewer gkKW: Major Issue 6, 7**
>
> **Major Issue 6:** The normalization layer highly impacts the performance of CNNs. The authors only used the default BatchNorm layer for CNNs in their experiments. It would be great if the authors provide some initial results using other normalization layers such as GroupNorm or recently proposed KernelNorm.
> **Response:**
>
> We agree that the normalization layer could have an impact on performance.  However, the goal of this study was to carefully analyze (and isolate) the impact of pooling operators on shift invariance.  While layer normalization is not the focus of the work, we have experimented on different alternatives and show results in Tables below.
> Tab F (batch size 32), G (batch size 256) contain results on CIFAR-10 with a ResNet-18 backbone with TIPS and MaxPool pooling with Layer Norm, Batch Norm, Group Norm, and Kernel Norm. We observe that usage of normalization layers leads to mixed results – this points to normalization not being a major factor for shift invariance.  However, like we mentioned before, studying the impact of layer normalization is not the focus of our work. However, Tab F, G reveal that using TIPS instead of baseline MaxPool improves shift invariance regardless of layer normalization choice.
>
> | ResNet-18       | CIFAR-10       | Standard Shift   |                    |                    | Circular Shift   |                    |
> |-----------------|----------------|------------------|--------------------|--------------------|------------------|--------------------|
> | **Batch size = 32** | **Accuracy**       | **Consistency**      | **Fidelity**           | **Consistency**        | **Fidelity**         |
> | Batch Norm [4]  | 96.02/91.43    | **98.61**/87.43      | **94.69**/79.94        | **100.00**/90.18       | 96.02/82.45      |
> | Layer Norm [3]  | 93.43/92.25    | 97.34/**89.37**      | 90.95/**89.77**        | **100.00**/90.61       | 93.43/83.60      |
> | Group Norm [5]  | 94.79/89.04    | 95.82/82.37      | 90.84/73.34        | **100.00**/**93.59**       | 94.79/83.33      |
> | Kernel Norm [6] | **96.18**/**95.72**    | 98.07/86.12      | 94.31/82.43        | **100.00**/90.81       | **96.18**/**86.95**      |
> ||
> **Table F:** Inspecting the influence of different layer normalization strategies on CIFAR-10 with TIPS pooling using a ResNet-18 backbone (batch size of 32). All results are reported as TIPS/MaxPOOL.
>
> | ResNet-18       | CIFAR-10       | Standard Shift   |                    |                    | Circular Shift   |                    |
> |-----------------|----------------|------------------|--------------------|--------------------|------------------|--------------------|
> | **Batch size = 256** | **Accuracy**       | **Consistency**      | **Fidelity**           | **Consistency**        | **Fidelity**         |
> | Batch Norm [4]  | **94.71/90.87** | 97.29/**88.29**      | 92.15/80.21        | **100.00**/84.91       | **94.71/76.89**  |
> | Layer Norm [3]  | 94.67/91.19    | 96.43/82.13      | 91.29/74.91        | **100.00**/89.02       | 94.67/81.24      |
> | Group Norm [5]  | 94.14/94.02    | 96.80/84.32      | 91.14/79.38        | **100.00**/92.30       | 94.14/86.69      |
> | Kernel Norm [6] | 94.58/**94.58** | **97.44**/86.36  | **92.16**/**81.69**    | **100.00**/**93.47**       | 94.58/**88.43**  |
> ||
> **Table G:** Inspecting the influence of different layer normalization strategies on CIFAR-10 with TIPS pooling using a ResNet-18 backbone (batch size of 256). All results are reported as TIPS/MaxPOOL.
>
> **Major Issue 7:** The difference between the previous approaches such as APS and LPS and the proposed approach of TIPS is unclear..
> **Response:**
>
> Although TIPS, APS, and LPS all use polyphase decomposition for spatial downsampling, they differ in how the pooled features are sampled from the decomposed polyphase components.
> - APS simply samples the polyphase component that contains the maximum energy using $\ell_{p}$ norm.
> - LPS learns to sample from these polyphase components using a shared convolution layer and gumble softmax.
> - In TIPS, the shared small convolution layer differs in design (Fig 3) from LPS. In TIPS layers, we use convolution kernels, Global Average Pooling (GAP) layer and softmax activation that learns mixing coefficients (eqn 2) to sample polyphase components avoiding the sensitivity to gumble softmax temperature.

---

> ### Author Response · Authors · 2024-06-17
> **Response #5 to reviewer gkKW: Minor Issue 1, 2, 3, 4.**
>
> **Minor Issue 1:** Figure 2 and the corresponding description are confusing: It seems that x and x' are identical inputs, i.e. transformation g is an identity mapping. However, they need to be different in order to see how the model output changes.
> **Response:**
>
> Figure 2 is for illustration purposes and the intent is to demonstrate that x' is a shifted version of x in the input space where g is the transformation function that causes spatial shift (x to x'). We acknowledge that x and x' look identical and we will fix this issue in the revision.
>
> **Minor Issue 2:** Section 3: "A pooling layer with stride s, downsamples X into ...". It should be specified that kernel size and stride values are identical and padding is zero.
> **Response:**
>
> We will address these changes in the revised version.
>
> **Minor Issue 3:** Last sentence of page 4: "using the using Kaiming normal approach" -> "using the Kaiming normal approach".
> **Response:**
>
> We will address these changes in the revised version.
>
> **Minor Issue 4:** The authors chose ϵ = 0.4 and α = 0.35 in the experiments. They should clarify how those values for ϵ and α have been obtained. Using parameter tuning?
> **Response:**
>
> The values of epsilon and alpha were obtained using hyperparameter search on CIFAR-10.
> Note that we only run hyperparameter tuning on CIFAR-10 (a small dataset) and then use the same hyperparameters for other image classification and semantic segmentation benchmarks without any hyperparameter search on those datasets.
> We chose ϵ = 0.4 because introducing $L_{undo}$ after 40% of the training duration yields the best performance – this observation is shown in Fig 8.
>
> **Overall:**
> We thank the reviewer for the insightful questions. We will add and discuss all of this additional results and analysis in the appendix of the manuscript.
>
> **References:**
>
> [1] Huang, Gao, et al. "Densely connected convolutional networks." Proceedings of the IEEE conference on computer vision and pattern recognition. 2017.
>
> [2] Tan, Mingxing, and Quoc Le. "Efficientnet: Rethinking model scaling for convolutional neural networks." International conference on machine learning. PMLR, 2019.
>
> [3] Ba, Jimmy Lei, Jamie Ryan Kiros, and Geoffrey E. Hinton. "Layer normalization." arXiv preprint arXiv:1607.06450 (2016).
>
> [4] Ioffe et al., "Batch normalization: Accelerating deep network training by reducing internal covariate shift", ICML, 2015.
>
> [5] Wu et al., "Group Normalization", ECCV, 2018.
>
> [6] Nasirigerdeh et al., "Kernel Normalized Convolutional Networks", TMLR, 2024.

---

### Review · Reviewer_MoRV · 2024-06-06

**Summary Of Contributions:**

The paper mainly introduces Translation Invariant Polyphase Sampling (TIPS), a learnable pooling operator designed to enhance the shift invariance of convolutional neural networks (CNNs) by reducing maximum-sampling bias (MSB). TIPS, along with two regularization techniques, significantly improves performance in image classification and semantic segmentation tasks, offering consistent gains in accuracy, shift consistency, and robustness. Extensive experiments show the effectiveness of the proposed training framework.

**Audience:**

Yes

**Broader Impact Concerns:**

I haven't seen any obviously ethical concerns of this work.

**Claims And Evidence:**

Yes

**Requested Changes:**

I hope the authors can improve the figure, writing, and experiments as raised in the Weakness part.

Overall, although I have some concerns about this work, I believe this paper presents a solid analysis of the correlation between shift invariance and model performance. The proposed method also works well, and many concerns won't be the main reason for me not to recommend acceptance.

**Strengths And Weaknesses:**

Strengths:

1: This paper provides an insightful metric to evaluate shift invariance (MSB) and systematically studies the correlation between MSB and model performance. The analysis is impressive.

2: The authors propose a new learnable pooling method called TIPS, designed to improve shift invariance in the widely-used pooling operation. TIPS can be integrated into any CNN and is trained end-to-end with reduced computational cost on CUDA memory (compared with GAP operation).

3: The overall training framework achieves state-of-the-art performance in terms of accuracy and various measures of shift invariance, outperforming previous methods. Meanwhile, the proposed method exhibits greater robustness to adversarial attacks compared to other shift-invariant pooling operators.

Weaknesses:

1: **Experiments on the object detection task.** The authors heavily discuss the performance of image classification and segmentation but ignore the object detection task, which is also an important object-centric task to verify the effectiveness of the shift-invariance.

2: **Does the "strongest signal" (Introduction, paragraph 2) mean the biggest value?** I agree that the feature activation maximum does influence the pooled value. However, does it mean simply sampling the biggest value leads to better performance? More clarification improves the reading.

3: I encounter some confusion in **Figure 4.** I have tried my best to understand Figure 4, but I still don't understand some designs: 1) Is the TIPS separate from the convolution block? 2) Why are there two arrows pointing in opposite directions (like <- ... ->)? I encourage the authors to refine this figure.

4: Is this a **wrongly placed paragraph?** The last paragraph on page 4 doesn't correspond to Eq. (5). Should it be placed after Eq. (4)?

5: Discussion and comparison of the **decomposition operation** in the TIPS layer with previous work. To the best of my knowledge, the decomposition operation that slices the feature map into 4 parts is very similar to previous works, e.g., focus layer in Yolo v5[1], dilated convolution[2], and dilated attention[3]. Could the authors provide more comparisons?

[1] https://github.com/ultralytics/yolov5.

[2] Yu, Fisher, and Vladlen Koltun. "Multi-scale context aggregation by dilated convolutions." arXiv preprint arXiv:1511.07122 (2015).

[3] Jiao, Jiayu, et al. "Dilateformer: Multi-scale dilated transformer for visual recognition." IEEE Transactions on Multimedia 25 (2023): 8906-8919.

---

> ### Author Response · Authors · 2024-06-17
> **Response #1 to reviewer MoRV: Weakness 1**
>
> **Weaknesses 1:** The authors heavily discuss shift invariance/euivariance performance of image classification and segmentation but ignore important object-centric task - object detection.
> **Response:**
>
> Although recent work on shift invariance (BlurPool, APS, LPS, etc.) only focus on image classification and semantic/instance segmentation, we thank the reviewer for this suggestion, and in the response phase we have studied the efficacy of different pooling methods including our proposed TIPS pooling in reducing shift equivariance.
> Following the training procedure and hyperparameters in the only study we found on shift equivariance on object detection [1], we train MS-COCO dataset on RetinaNet [2] and FasterRCNN [3].
> We train BlurPool, APS, LPS, TIPS (ϵ = 0.4, α = 0.35) with LPF-5 and report performance on MS-COCO validation set in Tab H, I, J, K.
> To evaluate shift equivariance we also follow the algorithm for greedy approximation of AP Variations proposed by [1].
> However, [1] only evaluates on standard shift invariance and to evaluate on circular shift invariance we apply the following change to the algorithm in [1] .
> We only consider the overlapping region between the shifted, non shifted images analogous to our semantic segmentation experiments (section 5.2).
> In Tab H, I, J, K we report $AP$, $\Delta AP$ (=best $AP$ - worst $AP$), $AP_{50}$ , $\Delta AP_{50}$ (=best $AP_{50}$ - worst $AP_{50}$) for both standard and circular shift invariance. Where higher value of $AP$ and $AP_{50}$ denotes superior object detection performance and lower value of $\Delta AP$ and $ AP_{50}$ denotes superior shift equivariant object detection performance.
> Tab H, I, J, K demonstrates that TIPS consistently achieves lower $\Delta AP_{50}$  and higher $AP_{50}$  which demonstrates that TIPS is also effective for shift equivariant object detection in comparison to existing pooling methods.
>
> | Pooling Method | $AP$   | worst $AP$ / best $AP$ | $\Delta AP$ | $AP_{50}$ | worst $AP_{50}$ / best $AP_{50}$ | $\Delta AP_{50}$ |
> |----------------|------|-----------------|-----|-------|--------------------|--------|
> | MaxPool        | 36.5 | 35.3 / 37.5     | 2.2 | 56.7  | 53.9 / 59.0        | 5.1    |
> | BlurPool       | 35.2 | 34.3 / 35.7     | 1.4 | 55.1  | 51.4 / 58.6        | 3.4    |
> | APS            | 36.8 | 36.2 / 37.4     | 1.2 | 56.7  | 53.9 / 58.5        | 4.6    |
> | LPS            | 36.9 | 36.5 / 37.6     | 1.1 | **56.8**  | 54.2 / 58.0        | 3.8    |
> | TIPS           | **37.0** | 36.5 / 37.4 | **0.9** | **56.8** | 54.9 / 57.6 | **2.7** |
> **Table H:** Standard Shift Equivariance for Object Detection on MS-COCO Validation Set with RetinaNet [2].
>
> | Pooling Method | $AP$   | worst $AP$ / best $AP$ | $\Delta AP$ | $AP_{50}$ | worst $AP_{50}$ / best $AP_{50}$ | $\Delta AP_{50}$ |
> |----------------|------|-----------------|-----|-------|--------------------|--------|
> | MaxPool        | 36.4 | 35.6 / 37.9     | 2.3 | 56.7  | 53.8 / 59.2        | 5.4    |
> | BlurPool       | 36.3 | 35.8 / 37.4     | 1.6 | 56.3  | 53.1 / 58.5        | 5.4    |
> | APS            | 37.4 | 35.7 / 37.8     | 2.1 | 56.9  | 54.0 / 58.5        | 4.5    |
> | LPS            | 37.5 | 35.8 / 37.7     | **1.9** | 57.2  | 54.3 / 58.5        | 4.2    |
> | TIPS           | **38** | 36.2 / 38.2  | 2.0 | **57.4** | 54.7 / 58.4 | **3.7** |
> **Table I:** Circular Shift Equivariance for Object Detection on MS-COCO Validation Set with RetinaNet [2].
>
> | Pooling Method | $AP$   | worst $AP$ / best $AP$ | $\Delta AP$ | $AP_{50}$ | worst $AP_{50}$ / best $AP_{50}$ | $\Delta AP_{50}$ |
> |----------------|------|-----------------|-----|-------|--------------------|--------|
> | MaxPool        | 37.6 | 36.5 / 39.4     | 2.9 | 59.0  | 55.7 / 62.1        | 6.4    |
> | BlurPool       | 37.8 | 36.6 / 38.6     | 2.0 | 58.7  | 56.3 / 60.9        | 4.6    |
> | APS            | 38.5 | 37.3 / 39.1     | 1.8 | 59.4  | 57.6 / 59.8        | 2.2    |
> | LPS            | 38.5 | 37.4 / 39.1     | 1.7 | 60.3  | 59.4 / 60.5        | 1.1    |
> | TIPS           | **38.6** | 37.6 / 38.9  | **1.3** | **60.5** | 59.7 / 60.7 | **1.0** |
> **Table J:** Standard Shift Equivariance for Object Detection on MS-COCO Validation Set with FasterRCNN [3].
>
> | Pooling Method | $AP$   | worst $AP$ / best $AP$ | $\Delta AP$ | $AP_{50}$ | worst $AP_{50}$ / best $AP_{50}$ | $\Delta AP_{50}$ |
> |----------------|------|-----------------|-----|-------|--------------------|--------|
> | MaxPool        | 37.8 | 36.2 / 39.6     | 3.4 | 59.3  | 56.2 / 60.5        | 4.3    |
> | BlurPool       | 38.3 | 36.5 / 40.2     | 3.7 | 59.5  | 58.3 / 59.6        | 1.3    |
> | APS            | 38.3 | 36.5 / 40.1     | 3.6 | 59.6  | 58.8 / 59.9        | 1.1    |
> | LPS            | 38.7 | 37.1 / 39.5     | **2.4** | 59.9  | 58.9 / 60.0        | 1.1    |
> | TIPS           | **38.9** | 37.5 / 40.0  | 2.5 | **60.1** | 59.4 / 60.2 | **0.8** |
> **Table K:** Circular Shift Equivariance for Object Detection on MS-COCO Validation Set with FasterRCNN [3].

---

> ### Author Response · Authors · 2024-06-17
> **Response #2 to reviewer MoRV: Weakness 2, 3, 4, 5**
>
> **Weaknesses 2:** Does the "strongest signal" (Introduction, paragraph 2) mean the biggest value? I agree that the feature activation maximum does influence the pooled value. However, does it mean simply sampling the biggest value leads to better performance?
> **Response:**
>
> By “strongest signal” we refer to downsampling the largest value or the maximum feature activation in the pooling window. Existing pooling methods such as maxpool, average pool, BlurPool, APS, LPS work well for downstream tasks on unshifted inputs however, are subject to both inferior shift invariance performance (especially on standard shift) and the aforementioned biased to pool strong signals. This observation motivates us to study- while the strategy of “strongest signal” works well for unshifted inputs, does this affect shift invariance when CNNs are tested on spatially shifted inputs?
>
> **Weaknesses 3:** I encounter some confusion in Figure 4. I have tried my best to understand Figure 4, but I still don't understand some designs: 1) Is the TIPS separate from the convolution block? 2) Why are there two arrows pointing in opposite directions (like <- ... ->)? I encourage the authors to refine this figure
> **Response:**
>
> Fig 4 illustrates the end-to-end (from input to prediction with all the objective functions to optimize) pipeline of training TIPS with downstream task loss  $L_{task}$ and our proposed regularizations: $L_{undo}$,  $L_{FM}$.
> Bidirectional arrows depict feedforward and backpropagate signal flow through the network while unidirectional arrows depict feedforward signal propagations only.
> Pooling layers always follow activated convolution features, here in Fig 4, an input image I first goes through a set of convolution and activation layers and we call these convolution feature maps $X$ which is input to the TIPS pooling layer.
> And one can repeat stacks of {convolution, activation, pooling} any number of times depending on the design of CNN architecture until reaching the prediction layers of the network.
> Now, $L_{task}$ and $L_{FM}$ take the final prediction and target as input.
> And, $L_{undo}$ is a MSE loss between $X^t$ which is a frozen transformation (standard shift) of $X$ (input to TIPS) and $\psi(X)$ which is an intermediate convolution feature map within the TIPS layer (depicted in Fig 3).
>
> **Weaknesses 4:** Is this a wrongly placed paragraph? The last paragraph on page 4 doesn't correspond to Eq. (5). Should it be placed after Eq. (4)?
> **Response:**
>
> The last paragraph on page 4 does contain eqn 5, i.e. the sentence immediately preceding eqn 5 introduces the motivation and background for this equation.
>
> **Weaknesses 5:** Discussion and comparison of the decomposition operation in the TIPS layer with previous work. To the best of my knowledge, the decomposition operation that slices the feature map into 4 parts is very similar to previous works, e.g., focus layer in Yolo v5[1], dilated convolution[2], and dilated attention[3].
> **Response:**
>
> Within TIPS layers, we use polyphase decomposition which is comparable to dilated convolution [4] and dilated attention [5] where the stride and dilation rates are identical (Fig 3).
> Usage of strided convolution in the above convolution operations can also be used for spatial downsampling, however strided convolutions are still shift invariant [6].
> The slicing operation in polyphase decomposition is also identical to that of parallel grid pooling [7], focus layer in YOLOv5 [8].
> However, we learn to sample from these polyphase decompositions in the channel dimension while [7,8] stack these decompositions in the channel space and then uses group convolution to downsample across the channel dimension.
>
> **Overall:**
> We thank the reviewer for the insightful questions. We will add and discuss all of this additional results and analysis in the appendix of the manuscript.
>
> **References**
>
> [1] Manfredi, Marco, and Yu Wang. "Shift equivariance in object detection." Computer Vision–ECCV 2020 Workshops: Glasgow, UK, August 23–28, 2020, Proceedings, Part VI 16. Springer International Publishing, 2020.
>
> [2] Lin, Tsung-Yi, et al. "Focal loss for dense object detection." Proceedings of the IEEE international conference on computer vision. 2017.
>
> [3] Ren, Shaoqing, et al. "Faster r-cnn: Towards real-time object detection with region proposal networks." Advances in neural information processing systems 28 (2015).
>
> [4] Yu, Fisher, and Vladlen Koltun. "Multi-scale context aggregation by dilated convolutions." arXiv preprint arXiv:1511.07122 (2015).
> [5] Jiao, Jiayu, et al. "Dilateformer: Multi-scale dilated transformer for visual recognition." IEEE Transactions on Multimedia 25 (2023): 8906-8919.
>
> [6] Zhang, Richard. "Making convolutional networks shift-invariant again." International conference on machine learning. PMLR, 2019.
>
> [7] Takeki, Akito, et al. "Parallel grid pooling for data augmentation." arXiv preprint arXiv:1803.11370 (2018).
>
> [8] https://github.com/ultralytics/yolov5.

---

> ### Comment · Reviewer_MoRV · 2024-07-09
>
> Thanks for the authors' rebuttal. It addresses all of my concerns well, and I have decided to recommend this paper for acceptance.

---

### Review · Reviewer_LsJC · 2024-06-07

**Summary Of Contributions:**

The paper addresses the issue of shift-invariance in convolutional architectures. It introduces Maximum-Sampling Bias (MSB) as a metric to measure the bias of pooling layers in propagating input signals based on their magnitude. The paper demonstrates that MSB is anticorrelated with both shift-invariance and the network’s performance. To minimize MSB and enhance shift-invariance, it proposes Translation Invariance Polyphase Sampling (TIPS), a novel learnable pooling method. TIPS results in improved performance across image classification and segmentation benchmarks. Additionally, TIPS improves robustness against adversarial attacks and common corruptions.

**Audience:**

Yes

**Claims And Evidence:**

Yes

**Requested Changes:**

The paper presents a comprehensive and extensive analysis of MSB and TIPS on ResNet arichitectures (and UNets with ResNet backbones). However, to result more convincing, I think it would benefit from considering more architectures, as mentioned above.

Additionally, there are small presentation issues that should be easy to fix.

**Strengths And Weaknesses:**

**Strenghts**

The paper tackles an interesting problem, improving shift-invariance in image classifiers. It introduces new measures to quantify issues in pooling operations and proposes a new method that outperforms existing baselines. The analysis is well-supported by empirical experiments and the results, particularly for ResNet architectures, are convincing. Overall, the paper is well-written (see below for improvements).

**Weaknesses**

- The paper claims to evaluate 576 models across different architectures. Yet, upon closer inspection of Appendix A, all models but one are ResNets with varying number of layers. Would the same shift invariance and performance improvements apply to other convolutional architectures such as ConvNeXt, EfficientNet, or VGG?
- Sec. 6.6 on the appplicability of TIPS to ViTs states that ViTs are also not shift invariant and TIPS outperforms ViTs. Yet, I think it is important to clarify that even though CNNs with TIPS achieve a stronger invariance to shifts, the performance of ViTs (significantly) outperforms CNNs with TIPS (see Tab. 3).
- Some parts of the paper lack clarity, as some terminology is not properly introduced. For example, MSB should be defined and briefly explained in the abstract and introduction, along with non-standard metrics like shift consistency and fidelity (if mentioned).
- I am not sure I understand the part of the introduction where the authors write that although convolution operations are shift **equivariant**, pooling and strides break shift **invariance**.
- From Sec. 3.2, it is not clear why the losses are used in different training phases.
- Tables often do not report best results in bold, making it hard to parse them (given also the small font size)

*Minor formatting issues*: The font of bold math symbols is not consistent throughout the manuscript (e.g., the notation of the set of real numbers at the beginning of Sec. 3).

---

> ### Author Response · Authors · 2024-06-17
> **Response #1 to reviewer LsJC: Weakness 1 (to be continued to Response #2 to reviewer LsJC)**
>
> **Weaknesses 1:** The paper claims to evaluate 576 models across different architectures. Yet, upon closer inspection of Appendix A, all models but one are ResNets with varying number of layers. Would the same shift invariance and performance improvements apply to other convolutional architectures such as ConvNeXt, EfficientNet, or VGG?
> **Response:**
>
> For clarity, we evaluate downstream task performance, shift consistency and fidelity on 576 different configurations. These configurations as shown in Tab 8 are achieved with a combination of 4 unique CNN models, 4 unique datasets, varying (4) number of pooling layers, 9 unique pooling methods (4x4x4x9=576).  To consider a wide variety of CNN design strategies in our correlation study, we use ResNets which has architectural choices such as residual / skip connections with varying depth and MobileNet which contains group (depthwise and separable) convolutions. However, to make our correlation study more robust we consider other configurations such as number of pooling layers, different datasets with varying magnitude of image resolution,  number of classes etc. Moreover, we train and test all of these 576 configurations which is computationally expensive while using models such as VGG [6],  ConvNext [7].
>
> However, we thank the reviewer for this suggestion, and in the response phase we have studied the efficacy of CNN architectures other than ResNets with different pooling methods including our proposed TIPS pooling in reducing shift equivariance. Tab B, C, D, E contain results from DenseNet-BC (k=24) [4] and EfficientNet-B7 [5] on CIFAR-10 and CIFAR-100 with different pooling methods including TIPS. We use low pass filters (LPF-5) for antialiasing and hyperparameters used in DenseNet and EfficientNet respectively. Our observation (Tab 1, 2, 3) with ResNet backbones which depicts that TIPS improves accuracy, shift consistency and fidelity holds true for DenseNet and EfficientNet.
>
> | **DenseNet-BC (k=24) [4]** | CIFAR-10        | Standard |  Shift                 | Circular |   Shift                |
> |----------------------------|-----------------|----------------|-------------------|----------------|-------------------|
> | **Pooling Method**         | **Accuracy**    | **Consistency**     | **Fidelity**      | **Consistency**     | **Fidelity**      |
> | MaxPool                    | 96.37           | 90.02          | 86.72             | 92.41          | 89.05             |
> | BlurPool                   | 96.74           | 92.57          | 89.51             | 94.07          | 90.96             |
> | APS                        | 97.27           | 93.31          | 90.82             | **100.00**         | 97.27             |
> | LPS                        | 97.12           | 94.36          | 91.62             | **100.00**         | 97.12             |
> | TIPS (ours)                | **97.43**       | **96.71**      | **94.19**         | **100.00**     | **97.43**         |
> ||
> **Table B:**  Image classification performance on CIFAR-10 with DenseNet-BC (k=24).
>
> | **DenseNet-BC (k=24) [4]** | CIFAR-100        | Standard |  Shift                 | Circular |   Shift                |
> |----------------------------|-----------------|---------------------|-------------------|---------------------|-------------------|
> | **Pooling Method**         | **Accuracy**    | **Consistency**     | **Fidelity**      | **Consistency**     | **Fidelity**      |
> | MaxPool                    | 80.35           | 92.72               | 74.58             | 89.63               | 72.02             |
> | BlurPool                   | 81.92           | 93.41               | 76.54             | 90.86               | 74.47             |
> | APS                        | 83.29           | 94.27               | 78.52             | **100.00**              | 83.29             |
> | LPS                        | 83.87           | 95.31               | 79.91             | **100.00**              | 83.87             |
> | TIPS (ours)                | **83.89**       | **96.17**           | **80.64**         | **100.00**          | **83.89**         |
> ||
> **Table C:**  Image classification performance on CIFAR-100 with DenseNet-BC (k=24).

---

> ### Author Response · Authors · 2024-06-17
> **Response #2 to reviewer LsJC: Weakness 1 (continued from Response #1 to reviewer LsJC), 2, 3**
>
> **Weaknesses 1:** Continued from the last comment…
> **Response:**
>
> | **EfficientNet-B7 [5]**          | CIFAR-10        | Standard |  Shift                 | Circular |   Shift                |
> |----------------------------|-----------------|---------------------|-------------------|---------------------|-------------------|
> | **Pooling Method**         | **Accuracy**    | **Consistency**     | **Fidelity**      | **Consistency**     | **Fidelity**      |
> | **MaxPool**                      | 98.90           | 89.14               | 88.13             | 92.19               | 91.22             |
> | **BlurPool**                     | 98.90           | 91.06               | 90.07             | 92.37               | 91.39             |
> | **APS**                          | 98.53           | 92.30               | 90.95             | **100.00**              | 98.53             |
> | **LPS**                          | **98.93**       | 93.47               | 92.44             | **100.00**              | **98.93**         |
> | **TIPS (ours)**                  | **98.93**       | **93.67**           | **92.67**         | **100.00**          | **98.93**         |
> ||
> **Table D:**  Image classification performance on CIFAR-10 with EfficientNet-B7.
>
> | **EfficientNet-B7 [5]**          | CIFAR-100       | Standard |  Shift                 | Circular |   Shift                |
> |----------------------------|-----------------|---------------------|-------------------|---------------------|-------------------|
> | **Pooling Method**         | **Accuracy**    | **Consistency**     | **Fidelity**      | **Consistency**     | **Fidelity**      |
> | MaxPool                    | 91.72           | 93.51               | 85.76             | 90.25               | 82.78             |
> | BlurPool                   | 92.17           | 94.68               | 87.27             | 92.39               | 85.20             |
> | APS                        | 92.18           | 94.57               | 87.31             | **100.00**              | 92.18             |
> | LPS                        | 93.50           | 95.81               | 89.60             | **100.00**              | 93.50             |
> | TIPS (ours)                | **93.71**       | **96.04**           | **89.99**         | **100.00**          | **93.71**         |
> ||
> **Table E:**  Image classification performance on CIFAR-100 with EfficientNet-B7.
>
> **Weaknesses 2:** Sec. 6.6 on the appplicability of TIPS to ViTs states that ViTs are also not shift invariant and TIPS outperforms ViTs. Yet, I think it is important to clarify that even though CNNs with TIPS achieve a stronger invariance to shifts, the performance of ViTs (significantly) outperforms CNNs with TIPS (see Tab. 3).
> **Response:**
>
> We agree that, while CNNs with TIPS achieve higher shift consistency, ViTs pretrained on ImageNet-21k outperform CNNs by a comparable margin on downstream tasks. Therefore, while large-scale pre-training with ViT models has no implications on shift invariace, ViTs are significantly superior on downstream tasks on unshifted inputs. In Tab L, we observe the following relative performance improvement (%) on image classification accuracy achieved by the most shift invariant ViT compared to the most shift invariant CNN model on 6 image classification datasets.
>
> | **Relative classification accuracy improvement by the most shift invariant ViT in comparison to most shift invariant CNN in terms of** | **CIFAR-10** | **CIFAR-100** | **Food-101** | **Oxford-102** | **Tiny ImageNet** | **ImageNet** |
> |-----------------------------------------------------------------------------------------------------------------|------------|--------------|------------|-------------|----------------|-----------|
> | **Standard Shift Consistency**                                                                                   | 3.30       | 1.56         | 1.11       | 1.75        | 4.41           | 4.67      |
> | **Circular Shift Consistency**                                                                                   | 3.23       | 1.56         | 0.48       | 2.11        | 4.92           | 4.67      |
> ||
> **Table L:**  Relative performance improvement (%) on image classification achieved by the most shift invariant ViT compared to the most shift invariant CNN model.
>
> **Weaknesses 3:** Some parts of the paper lack clarity, as some terminology is not properly introduced. For example, MSB should be defined and briefly explained in the abstract and introduction, along with non-standard metrics like shift consistency and fidelity (if mentioned).
> **Response:**
>
> We will address these changes in the revised version.

---

> ### Author Response · Authors · 2024-06-17
> **Response #3 to reviewer LsJC: Weakness 4, 5, 6, Minor formatting issues**
>
> **Weaknesses 4:** I am not sure I understand the part of the introduction where the authors write that although convolution operations are shift equivariant, pooling and strides break shift invariance.
> **Response:**
>
> Convolution operations are shift equivariant, i.e. a convolution filter that detects vertical edges will detect them in any spatial location of an image. Both pooling operators following convolution layers and strided convolutions are shown to break shift invaraiance [1,2,3], i.e. pooled feature maps of shifted and non shifted inputs are not identical which results in shift invariance for vision tasks such as image classification.
>
> **Weaknesses 5:** From Sec. 3.2, it is not clear why the losses are used in different training phases.
> **Response:**
>
> Downstream task loss $L_{down}$ and regularization to discourage known Failure Modes of shift invariance $L_{FM}$ are optimized throughout the entirety of the training phase.
> However, the proposed regularization to undo standard shift $L_{undo}$ is introduced to the optimization pipeline after a few epochs have been trained with only $L_{down}$ and $L_{FM}$, as shown in eqn 5.
> Optimizing with $L_{undo}$ from the very beginning hurts downstream task performance and with this observation (Fig 8) we optimize different losses in different training phases (eqn 5).
>
> **Weaknesses 6:** Tables often do not report best results in bold, making it hard to parse them (given also the small font size).
> **Response:**
>
> We will address these changes in the revised version.
>
> **Minor formatting issues:** The font of bold math symbols is not consistent throughout the manuscript (e.g., the notation of the set of real numbers at the beginning of Sec. 3).
> **Response:**
>
> We will address these changes in the revised version.
>
> **Overall:**
> We thank the reviewer for the insightful questions. We will add and discuss all of this additional results and analysis in the appendix of the manuscript.
>
> **References:**
>
> [1] Zhang, Richard. "Making convolutional networks shift-invariant again." International conference on machine learning. PMLR, 2019.
>
> [2] Chaman, Anadi, and Ivan Dokmanic. "Truly shift-invariant convolutional neural networks." Proceedings of the IEEE/CVF Conference on Computer Vision and Pattern Recognition. 2021.
>
> [3] Karras, Tero, et al. "Alias-free generative adversarial networks." Advances in neural information processing systems 34 (2021): 852-863.
>
> [4] Huang, Gao, et al. "Densely connected convolutional networks." Proceedings of the IEEE conference on computer vision and pattern recognition. 2017.
>
> [5] Tan, Mingxing, and Quoc Le. "Efficientnet: Rethinking model scaling for convolutional neural networks." International conference on machine learning. PMLR, 2019.
>
> [6] Simonyan, Karen, and Andrew Zisserman. "Very deep convolutional networks for large-scale image recognition." arXiv preprint arXiv:1409.1556 (2014).
>
> [7] Liu, Zhuang, et al. "A convnet for the 2020s." Proceedings of the IEEE/CVF conference on computer vision and pattern recognition. 2022.

---

### Author Response · Authors · 2024-07-05
**Checking for further questions from the reviewers**

Dear reviewers,

we thank you again for your valuable feedback. With our comments to each review, we have responded to all of the major and minor concerns or suggestions that were raised in the review, with additional experiments and results when necessary. Can you please acknowledge if you have received our responses and please let us know if you have any further questions or concerns?

---

> ### Comment · Reviewer_gkKW · 2024-07-12
> **Not fully addressed concerns**
>
> Dear authors,
>
> Thank you for your detailed response to the comments! I think my concerns regarding the following major issues have not been fully addressed:
>
> 1. "The accuracy results for the CIFAR-100 dataset seem too high. Based on previous studies, ResNets can achieve maximum accuracy of around 80% on CIFAR-100. The authors should provide a detailed description of their training procedure, or mention the baseline study they followed for training the models if that is the case."
>
> Could you please describe the training procedure and hyper-parameter tuning used in the experiments?
>
> By training procedure, I mean the optimizer, loss function, learning rate scheduler, how the base learning rate decayed during training, and etc. By hyper-parameter tuning, I mean which range of learning rates used, and what is the best learning rate in the experiments.
>
>
> 2. "The proposed method requires much more trainable parameters than many of the state-of-the-art pooling methods. For instance, TIPS based ResNet-34 has around 1 million more parameters than Max-pooling based counterpart according to Table-15. How can we ensure that improvement from TIPS is not because of more trainable parameters?"
>
> The provided results do not report the accuracy values!
>
> What are the accuracy values for LPS?
>
> In general, CIFAR-10 is an easy dataset for models like EfficientNet-B7!

---

> ### Author Response · Authors · 2024-07-13
> **Response to remaining concerns by Reviewer gkKW**
>
> We thank the reviewer for acknowledging our response. The following is our response to the concerns addressed above:
>
> 1. We use SGD optimizer with momentum 0.9, and weight decay 1e-4. The initial learning rate is 0.05, and the decay is 0.2. We train pre-activation ResNet-34 [1] on CIFAR-100 for 250 epochs using cosine annealing with patience of 10 epochs and step size of 50. Furthermore we use warm-up for large batch training (batch size of 64). We also augment images following strategies in [2, 3] for training. Moreover, our code is already available in supplementary material and also contains these details for reproducibility.
>
> 2. Accuracy, Standard and Circular shift consistency and fidelity for other pooling methods (BlurPool, APS, LPS) with DenseNet-BC (k=24) and EfficientNet-B7 can be observed in Tab B, C, D, E. We also report CIFAR-100 results in these tables under the aforementioned setup.
> Table A* (Table A + corresponding accuracy values) shows a comparison between MaxPool and TIPS using CNN architectures with different magnitudes of trainable parameters.
> We observe that heavier increase in trainable parameters by switching to bigger models such as ResNet-18 to EfficientNet-B7 is more effective in improving accuracy than shift consistency.
> However, a lesser increase in trainable parameters from switching MaxPool to TIPS is more effective in improving shift consistency than accuracy.
> The above observation depicts that adding more trainable parameters with TIPS is effective in improving shift consistency while adding more trainable parameters with larger architecture is effective in improving accuracy.
> This observation solidifies the contribution of the additional parameters in TIPS to improve shift consistency which cannot be achieved equivalently by just switching to larger models, even if the larger models contain more parameters than TIPS.
> This is observed across the board for different architectures and datasets.
>
> | Pooling         | MaxPool                   |           |               | TIPS                     |                  |        |
> |-----------------|---------------------------|--------------------------|--------------------------|--------------------------|--------------------------|--------------------------|
> | **CNN Model**       | **Parameters (M)**            | **Standard Shift Consistency** | **Accuracy** | **Parameters (M)**            | **Standard Shift Consistency** | **Accuracy** |
> | ResNet-18       | 11.9                      | 87.43              | 91.43 |         12.5                     |        98.65                    |     96.05  |
> | DenseNet-BC (k=24) [1] | 15.3                      | 90.02      | 96.37              | 16.9                     | 96.71                    |  97.43  |
> | ResNet-34       | 21.3                      | 88.93           | 93.44          | 22.3                     | 98.60                    |  94.39 |
> | EfficientNet-B7 [2] | 64.0                      | 89.14               |  98.90     | 67.3                     | 93.67                    | 98.93 |
> ||
>
> **Table A\*:** Inspecting how standard shift consistency in CIFAR-10 improves by increasing trainable parameters (bigger CNN models and switching pooling layers from MaxPool to TIPS).
>
>
> **References:**
> \
> **[1]** He, Kaiming, et al. "Identity mappings in deep residual networks." Computer Vision–ECCV 2016: 14th European Conference, Amsterdam, The Netherlands, October 11–14, 2016, Proceedings, Part IV 14. Springer International Publishing, 2016.
> \
> **[2]** Szegedy, Christian, et al. "Going deeper with convolutions." Proceedings of the IEEE conference on computer vision and pattern recognition. 2015.
> \
> **[3]** Dosovitskiy, Alexey, et al. "An image is worth 16x16 words: Transformers for image recognition at scale." arXiv preprint arXiv:2010.11929 (2020).

---

### Decision · Action_Editor_xSEQ · 2024-07-24

**Recommendation:** Reject

**Comment:**

Reviewer recommendations for this submission were mixed. The AE decided that the main concern of Reviewer gkKW  ("Strong reject" recommendation) has merit: The results in Table 1 for CIFAR-100 are surprisingly high: The authors claim that a resnet-34 with maxpooling achieves 88.38 accuracy - however this is not consistent with other published sources. The reviewer correctly states that "verified repositories like [1] report accuracy of around 80% for ResNets on CIFAR-100.". The AE agrees: [2] reports that a highly optimized, larger ResNet50 (A1) achieves 86.9% accuracy. Therefore, there seem to be some inaccuracies in at least one place in the submission's experimental evidence.

The criteria for acceptance at TMLR are whether "the claims made in the submission supported by accurate, convincing and clear evidence". In light of the experimental inaccuracies on CIFAR-100,  I am forced to recommend rejection.

Besides the inaccuracies mentioned above, the effectiveness of the proposed method seems to be limited to ResNet CNNs ("the proposed method, TIPS, provides no significant gain compared to the previous method, LPS, using architectures other than ResNet. E.g. For EfficientNet-B7 on CIFAR-100: LPS: 93.50%,  TIPS: 93.71%")

The AE recommends rejection of this manuscript in its current form.  However, reviewers also found the analysis "interesting" and that the paper "presents a comprehensive and extensive analysis of MSB and TIPS on ResNet architectures and UNets with ResNet backbones". The AE encourages the authors to look closely into their experimental evaluation and correct any discrepances.

[1] https://github.com/weiaicunzai/pytorch-cifar100

[2] ResNet strikes back: An improved training procedure in timm

**Audience:**

The findings would interest the subset of the TMLR audience that is interested in CNN invariances.

**Claims And Evidence:**

There seem to be inconsistencies in the experimental results (see comments below)

**Resubmission Of Major Revision:**

The authors may consider submitting a major revision at a later time.